# Color Conditional Generation with Sliced Wasserstein Guidance

**Alexander Lobashev**[1]    **Maria Larchenko** [2]    **Dmitry Guskov** [1, 3]

[1]Glam AI, San Francisco, USA

[2]Magicly AI, Dubai, UAE, [3]McGill University, Montreal, Canada

`{lobashevalexander, mariia.larchenko, guskov01dmitry}@gmail.com`

## Abstract

We propose SW-Guidance, a training-free approach for image generation conditioned on the color distribution of a reference image. While it is possible to generate an image with fixed colors by first creating an image from a text prompt and then applying a color style transfer method, this approach often results in semantically meaningless colors in the generated image. Our method solves this problem by modifying the sampling process of a diffusion model to incorporate the differentiable Sliced 1-Wasserstein distance between the color distribution of the generated image and the reference palette. Our method outperforms state-of-the-art techniques for color-conditional generation in terms of color similarity to the reference, producing images that not only match the reference colors but also maintain semantic coherence with the original text prompt. Our source code is available at `https://github.com/alobashev/sw-guidance`.

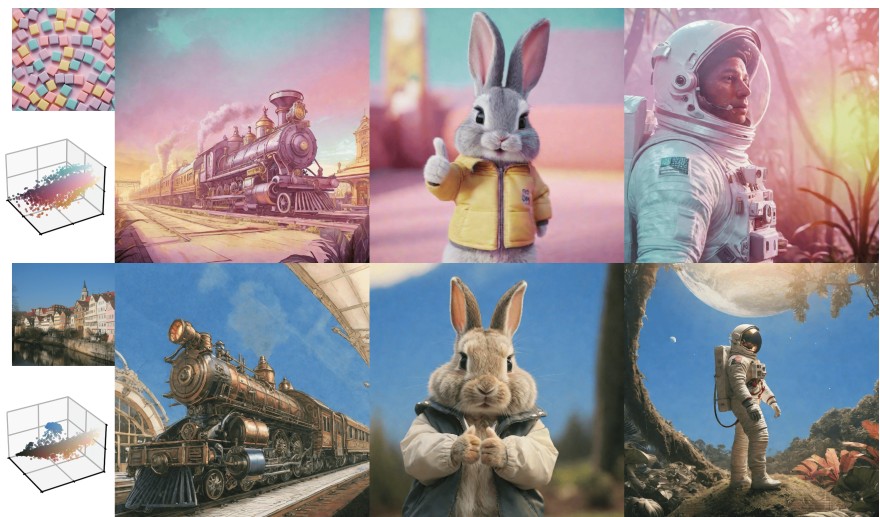

Figure 1: Color-conditional generation by Sliced Wasserstein guidance achieves unprecedented match with a reference color palette without transferring other stylistic features.

## 1 Introduction

To get a desired picture from text-to-image models we usually need a precise prompt and a bit of luck. However, natural language is not expressive enough to accurately describe colors, and even specific

terms such as "turquoise blue" yield varying tones. Moreover, prompt length constraints make full palette descriptions impractical. Using reference images for color styles addresses these limitations and establishes the color transfer problem, that is, applying a reference color style to a content image.

Color transfer is closely related to the artistic style transfer. Notably, artistic style is not linked to the depicted objects but is instead shared between patches of an image. This insight was utilized in the seminal work by Gatys *et al.* [1], where artistic style is defined as the distribution of activations in the VGG-19 network [2]. To match the artistic style between generated and reference images, Gatys *et al.* minimized the difference between the Gram matrices of activations from internal layers of VGG-19 (which is equivalent to matching the first two moments of the distributions of activations).

Strictly speaking, the style loss by Gatys *et al.* is not a proper distance in the space of probability distributions, as, for instance, Jensen-Shannon [3], Total variation and various Wasserstein distances [4]. Unfortunately, these metrics are hard to approximate in a differentiable fashion. To address the complications of Wasserstein metrics, a new family of metrics called Sliced Wasserstein (SW) distances was developed in 2012 [5, 6]. First, Sliced Wasserstein distances are differentiable. Second, they can be efficiently estimated from samples. Importantly, for bounded distributions, convergence of the Sliced Wasserstein distance implies the convergence of all moments. However, in high-dimensional spaces the sliced approach requires a large number of projections to accurately estimate the distance. To generalize the SW distance and enhance its performance in higher dimensions other its variants were proposed [7, 8, 9, 10, 11, 12, 13, 14].

Following Gatys *et al.*, various CNN-based color transfer methods were proposed, such as DPST [15], WCT [16], PhotoWCT [17], WCT2 [18], PhotoNAS [19], PhotoWCT2 [20], and DAST [21]. These algorithms can address the problem of color-conditional image generation, transferring reference colors to the image created by a text-to-image model.

Another way to achieve color conditioning is to control the generation process of a diffusion model [22, 23, 24]. This problem setting is broadly called the stylized image generation. The approaches for stylized generation could be categorized into three groups:

**Modification of weights**  The first group includes additive corrections of a model's weights, which require fine-tuning for every new style of images: Textual Inversion [25], DreamBooth [26], and LoRA [27]. The introduction of ControlNet [28] and T2I-Adapter [29] in 2023 enabled adjustments of weights in a single pass of a hyper-network. ControlNets and adapters are trained on fairly large paired datasets and cover tasks such as pose, depth, and edge conditioning.

**Modification of attention**  The examples of attention-related algorithms are IP-Adapter [30], StyleAdapter [31], StyleDrop [32], StyleAligned [33], InstantStyle [34, 35]. Training-free, they change attention output on each step and are effective for controlling structural and high-level features, such as painting style and composition, but do not target a color distribution separately.

**Modification of sampling**  The third way to impose a condition is to add a new term to the denoising process. The first work of this kind was classifier guidance [22], which requires a specific classifier trained on noisy data samples[1]. Diffusion Posterior Sampling (DPS) [36] addresses the main weakness of classifier guidance by replacing a noisy classifier with a composition of a predicted noiseless image and a classifier trained on clean data (i.e., any pre-trained one). Universal Diffusion Guidance [37] and FreeDoM [38] generalize the DPS approach by replacing the MSE loss used by DPS with a general distance function. These ideas were further developed in RB-Modulation [39].

In current approaches to stylized image generation style and color conditioning are often entangled, making it challenging to control these aspects independently. Our goal is to propose a way to condition solely on color and independently control the palette and general style of an image.

**Our Contributions**  This work makes the following key contributions:

- For the first time, we incorporate the differentiable Sliced Wasserstein distance and its generalizations into the conditioning of a diffusion model

- We achieve state-of-the-art results in a problem of color-specific conditional generation, without transferring unwanted textures or other stylistic features (see Fig. 1).

---

[1]This guided denoising procedure resembles the optimization process of Gatys *et al.*, which also generates an image from Gaussian noise by iterative denoising. In this case, the unconditional score function is equal to the gradient of a content loss, and the classifier guidance term corresponds to the gradient of a style loss.

## 2 Background

### 2.1 Conditioning Process in Diffusion Models

Diffusion models [40, 41] are a class of generative models that learn to iteratively denoise a data distribution. To describe the conditioning process in diffusion models, we use Bayes' rule to express the posterior distribution in terms of the gradient of the log-likelihood and the unconditional score:

$$\nabla_{x_t} \log p(x_t|y) = \nabla_{x_t} \log p(y|x_t) + \nabla_{x_t} \log p(x_t), \tag{1}$$

where $y$ represents the conditioning, and $x_t$ is the noisy sample at noise level $t$.

Diffusion Posterior Sampling (DPS) [36] introduced an approximation for the conditional likelihood based on a predicted noiseless sample, $\hat{x}_0 = \mathbb{E}(x_0|x_t)$, as

$$p(y|x_t) \approx p(y|\hat{x}_0(x_t)). \tag{2}$$

In the DPS approach, the authors considered the gradient of the log-likelihood as follows:

$$\nabla_{x_t} \log p(y|\hat{x}_0(x_t)) = -\frac{1}{\sigma^2} \nabla_{x_t} ||y - A(\hat{x}_0(x_t))||^2, \tag{3}$$

where $A$ is an operator, generally non-linear, that extracts the condition $y$ from the predicted noiseless sample $\hat{x}_0$, and $\sigma$ is a positive hyperparameter. For example, $A$ could extract the CLIP [42] embedding from $\hat{x}_0$, and $y$ could be a target prompt embedding.

Universal Diffusion Guidance [37] and FreeDoM [38] extend the DPS approximation by proposing a more general distance function $\mathcal{D}$ in the space of conditions $Y$. Specifically, for $y \in Y$, the gradient of the logarithm of the posterior distribution is given by:

$$\nabla_{x_t} \log p(y|\hat{x}_0(x_t)) = -\frac{1}{\sigma^2} \nabla_{x_t} \mathcal{D}\left(y, A(\hat{x}_0(x_t))\right). \tag{4}$$

This formulation is more flexible and lets $y$ and $A(\hat{x}_0(x_t))$ to be a more complicated objects than vectors in $\mathbb{R}^d$ as long as we can define a differentiable distance function between them. In the next section, we will define the Sliced Wasserstein distance as a suitable distance $\mathcal{D}$ between two probability measures.

### 2.2 Sliced Wasserstein distance

A classical formulation of color transfer problem is to align two probability distributions in the 3-dimensional RGB space. Specifically, the color distributions of a content image and a reference image can be represented as probability density functions, denoted by $\pi_0$ and $\pi_1$ respectively. The objective in guided diffusion models is to match the generated sample's probability density $\pi_0$ with the reference $\pi_1$.

Wasserstein distances, rooted in optimal transport theory, appear to be natural for this task as they measure the cost of transporting one probability distribution to match another [4]. The Wasserstein distance of order $p$ is

$$W_p(\pi_0, \pi_1) = \left( \inf_{\pi \in \Pi(\pi_0, \pi_1)} \int_{\mathcal{X}_0 \times \mathcal{X}_1} ||x - y||^p \, d\pi(x, y) \right)^{1/p}, \tag{5}$$

Calculating $W_p(\pi_0, \pi_1)$ for many samples can be computationally prohibitive, also a Wasserstein distance is hard to differentiate through, because its value is itself a result of an optimization procedure inf over all transport plans $\Pi(\pi_0, \pi_1)$, *i.e.* over all joint distributions with marginals $\pi_0$ and $\pi_1$.

To alleviate this issue, the Sliced Wasserstein (SW) distance was introduced [5], offering a more computationally tractable alternative by reducing high-dimensional distributions to one-dimensional projections where the Wasserstein distance can be computed more straightforwardly. The Sliced $p$-Wasserstein distance is defined as [5, 6]:

$$SW_p(\pi_0, \pi_1) = \left( \int_{\mathbb{S}^{d-1}} W_p^p(P_\theta \pi_0, P_\theta \pi_1) \, d\theta \right)^{1/p}, \tag{6}$$

where $\mathbb{S}^{d-1}$ is the unit sphere in $\mathbb{R}^d$ with $\int_{\mathbb{S}^{d-1}} d\theta = 1$, $P_\theta$ is a linear projection onto a one-dimensional subspace defined by $\theta$ and $W_p^p$ is an ordinary p-Wasserstein distance by Eq.5.

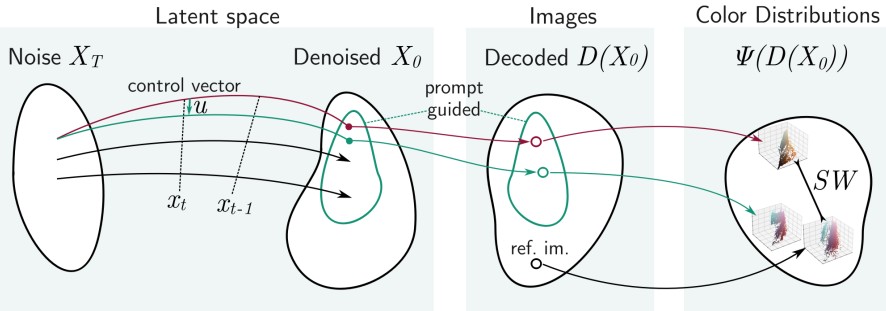

Figure 2: General scheme of the Slices Wasserstein Guidance for a latent diffusion model with decoder $D$ and feature extractor $\Psi$.

## 3   Method

Below we give a detailed description of SW-Guidance algorithm. We placed some necessary theoretical fact, Proposition 1 and Lemma 2, at the end of this section.

The general scheme of the algorithm is illustrated in Fig. 2. We denote $x_T, \dots, x_0$ as the latent states of our diffusion sampling, where $x_T$ is a sample from a normal distribution, $x_0$ is a noiseless sample and $\hat{x}_0(x_t)$ is a prediction of $x_0$ for given $x_t$. $D(x_0)$ is a decoded image from the latent space of diffusion to the real image domain. Lastly, $\Psi$ is a feature extractor, which in our case is the color distribution of an image in RGB color space.

---

**Algorithm 1** Color Conditional Generation with Sliced Wasserstein Guidance

---

1: **Initialize** latent vector $x_T \sim \mathcal{N}(0, I)$, set learning rate $\lambda_{\text{lr}}$, $y$ - samples from the reference color distribution
2: **for** $t = T$ to $1$ **do**
3:     $u \leftarrow \mathbf{0}$                                                                    ▷ Initialize control vector
4:     **for** $j = 1$ to $M$ **do**
5:         $x'_t \leftarrow x_t + u$
6:         Get prediction of last latent $\hat{x}_0 \leftarrow \text{DDIM}(t, x'_t)$
7:         Get $\hat{y}_0 \leftarrow \text{VAE}(\hat{x}_0)$                                        ▷ Decode latent to image
8:         **for** $k = 1$ to $K$ **do**                                                          ▷ Sliced Wasserstein
9:             Project samples on a random direction $\theta$
10:            Update loss $\mathcal{L} \leftarrow \mathcal{L} + \sum |\text{cdf}_{\hat{y}_0} - \text{cdf}_y|$
11:        **end for**
12:        Update control vector $u \leftarrow u - \lambda_{\text{lr}} \nabla_u \mathcal{L}(u)$
13:    **end for**
14:    Update latent $x^*_t \leftarrow x_t + u$
15:    Get denoised latent $x_{t-1} \leftarrow \text{DDIM}(t, x^*_t)$
16: **end for**

---

The proposed Algorithm 1 initializes a noise tensor $x_T$ sampled from a latent normal distribution. Over $T$ diffusion timesteps, the noise tensor is iteratively refined. Following each denoising step, a predicted result $\hat{x}_0$ is decoded to obtain an image $\hat{y}_0 = D(\hat{x}_0)$. To modulate guided diffusion within each timestep, we add an auxiliary control tensor $u$ following [39]. That is, $u$ is initialized with zeros and we set $x'_t = x_t + u$. Then we predict original sample $\hat{x}_0(x'_t) = \hat{x}_0(u)$ and compute gradient of the Sliced Wasserstein distance (SW) between the color distribution of a reference image $\pi_{\text{ref}}$ and the predicted $\hat{y}_0(u) = D(\hat{x}_0(u))$ with a respect to $u$

$$\mathcal{L}(u) = \text{SW}_1(\pi_{\hat{y}_0(u)}, \pi_{\text{ref}}), \tag{7}$$

The control vector $u$ is optimized over $M$ steps to shift the reverse diffusion process toward the reference's color distribution. This optimization accumulates gradients w.r.t $u$ $M$ times and minimizes the loss function $\mathcal{L}$, Eq.7. Let us note that by Lemma 2 the minimization of the loss $\mathcal{L}$ will lead

to a weak convergence of generated color distribution $\pi_{\hat{y}_0(u)}$ towards the reference $\pi_{\text{ref}}$ with the convergence of all moments.

The loss function can incorporate first two moments (mean $\mu$ and covariance $\sigma$) of $\pi_{\hat{y}_0(u)}$ and $\pi_{\text{ref}}$

$$\mathcal{L}(\hat{y}_0(u)) = (\mu_{\hat{y}_0} - \mu_{\text{ref}})^2 + (\sigma_{\hat{y}_0} - \sigma_{\text{ref}})^2 + \text{SW}(\pi_{\hat{y}_0}, \pi_{\text{ref}}), \tag{8}$$

The impact of adding the first two moments (see Fig. 5), along with other variants of the SW distance such as Generalized [8], Distributional [9], and Energy-Based [11] SW distances, is studied in the Experiments section.

Let $u^\star$ be a shift, obtained after M steps of Eq. 7 optimization. Then we set $x_t^\star = x_t + u^\star$ and perform usual DDIM [43] denoising step for $x_t^\star$ with classifier-free guidance to obtain $x_{t-1}$. Full algorithm for a latent diffusion model with classifier-free guidance is listed in the Appendix.

**Efficient Computation of Sliced Wasserstein**  Let $F_0$ and $F_1$ be two cumulative distribution functions of 1-dimensional probability distributions $\pi_0$ and $\pi_1$. Then the Wasserstein distance of order $p$ between $\pi_0$ and $\pi_1$ has a form (Rachev and Rüschendorf, 1998, Theorem 3.1.2 [44])

$$W_p(\pi_0, \pi_1) = \left( \int_0^1 \left| F_0^{-1}(y) - F_1^{-1}(y) \right|^p dy \right)^{\frac{1}{p}} \tag{9}$$

Formally, it involves differentiable estimation of inverse cumulative density functions. However, in the case of $p = 1$ the Proposition 1 allows us to replace the difference of inverse CDFs by absolution difference of CDFs, making it much easier to compute

$$W_1(\pi_0, \pi_1) = \int_{-\infty}^{\infty} |F_0(x) - F_1(x)| \, dx \tag{10}$$

Moreover, since all color distributions in RGB space have a compact support (unit cube), one can employ guarantees of Lemma 2, which in fact states a convergence of general p-Wasserstein distances given convergence of the 1-Wasserstein distance. These facts justify the selection of 1-Wasserstein instead of general $p$-Wasserstein.

**Differentiable Approximation of CDF**  We approximate the cumulative distribution function (CDF) by sorting samples from the distribution. Once the samples $\{x_i\}_{i=1}^n$ are sorted, the CDF can be directly obtained by assigning a rank to each sorted sample. For a given sample $x_i$, its rank (i.e., its position in the sorted array) divided by the total number of samples $n$ provides the CDF value at that point. If $\{x_{(i)}\}$ represents the sorted samples, the CDF at $x_{(i)}$ is given by:

$$\text{CDF}(x_{(i)}) = \frac{i}{n} \tag{11}$$

This sorting operation is differentiable, so the CDF is also differentiable. To achieve a good approximation of the true underlying CDF, a large number of samples $n$ is required.

**Theoretical Justification**  We need Proposition 1 for efficient sampling, as it allows one to avoid computing the inverse CDF.

**Proposition 1.** *Let $F$ and $G$ be two cumulative distribution functions. Then,*

$$\int_0^1 \left| F^{-1}(t) - G^{-1}(t) \right| \, dt = \int_{\mathbb{R}} |F(x) - G(x)| \, dx, \tag{12}$$

*where $F^{-1}$ and $G^{-1}$ are the quantile functions (inverse CDFs) of $F$ and $G$, respectively.*

Lemma 2 provides the theoretical foundation for our optimization procedure for multidimensional Borel probability measures $\mu_n$ and $\mu$ on $\mathbb{R}^d$.

**Lemma 2.** *Let $\mu_n$ and $\mu$ be Borel probability measures on the unit cube in $\mathbb{R}^d$. If the numerical sequence*

$$\lim_{n \to \infty} \text{SW}(\mu_n, \mu) = 0 \tag{13}$$

*then the sequence $\mu_n$ converges to $\mu$ weakly, and all moments of $\mu_n$ converge to the moments of $\mu$.*

Proofs for Proposition 1 and Lemma 2 are provided in the Appendix.

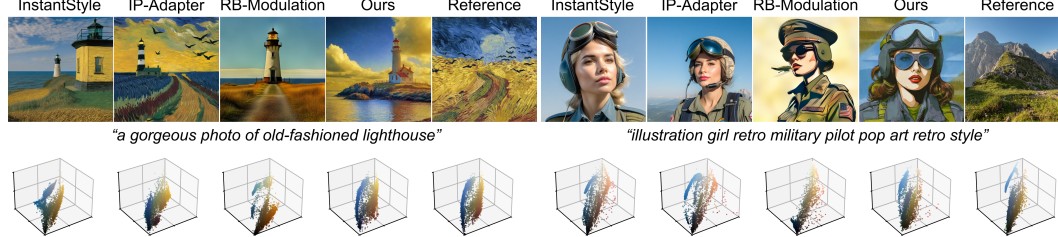

Figure 3: Comparison with stylized generation methods, SDXL. Other methods show weaker palette matching while transferring high-level features - such as brush strokes and wheat fields in example with lighthouses or photorealism in the second example.

Table 1: Quantitative evaluation, SDXL [46]. We measure palette similarity with 2-Wasserstein distance between the color distribution of the generated image and the reference image. CLIP-IQA and CLIP-T are quality and content scores. The color transfer methods [18, 19, 20, 52, 53, 54, 55, 56] are applied to the unconditional SDXL generations. Note, that SW-Guidance has the highest CLIP-T among other stylized generation algorithms [30, 34, 39]. For visual comparisons, see the Appendix.

| | 2-Wasserstein distance [4] ↓ | | Content scores | |
| --- | --- | --- | --- | --- |
| Algorithm | mean $\pm$ std of mean | | CLIP-IQA [51] ↑ | CLIP-T [42] ↑ |
| SW-Guidance SDXL (ours) | **0.0297 $\pm$ 0.0005** | | 0.285 $\pm$ 0.004 | 0.270 $\pm$ 0.002 |
| hm-mkl-hm [52] | 0.0543 $\pm$ 0.0011 | | 0.259 $\pm$ 0.003 | 0.277 $\pm$ 0.002 |
| hm [53] | 0.0856 $\pm$ 0.0016 | | 0.244 $\pm$ 0.003 | 0.282 $\pm$ 0.002 |
| PhotoWCT2 [20] | 0.1028 $\pm$ 0.0014 | | 0.225 $\pm$ 0.003 | 0.276 $\pm$ 0.002 |
| ModFlows [54] | 0.1125 $\pm$ 0.0016 | | 0.257 $\pm$ 0.003 | 0.282 $\pm$ 0.002 |
| MKL [55] | 0.1191 $\pm$ 0.0017 | | 0.238 $\pm$ 0.003 | 0.283 $\pm$ 0.002 |
| CT [56] | 0.1333 $\pm$ 0.0018 | | 0.230 $\pm$ 0.003 | 0.284 $\pm$ 0.002 |
| WCT2 [18] | 0.1347 $\pm$ 0.0017 | | 0.179 $\pm$ 0.002 | 0.288 $\pm$ 0.002 |
| PhotoNAS [19] | 0.1608 $\pm$ 0.0017 | | 0.167 $\pm$ 0.002 | 0.279 $\pm$ 0.002 |
| InstantStyle SDXL [34] | 0.1758 $\pm$ 0.0028 | | **0.332 $\pm$ 0.003** | 0.238 $\pm$ 0.002 |
| IP-Adapter SDXL [30] | 0.2193 $\pm$ 0.0032 | | 0.247 $\pm$ 0.002 | 0.214 $\pm$ 0.002 |
| Unconditional SDXL [48] | 0.3824 $\pm$ 0.0059 | | 0.239 $\pm$ 0.003 | **0.294 $\pm$ 0.002** |
| RB-Modulation [39] Stable Cascade | 0.3795 $\pm$ 0.0133 | | 0.323 $\pm$ 0.006 | 0.266 $\pm$ 0.003 |

## 4   Experiments

As a successor to Universal Diffusion Guidance [37], the proposed method is not tied to a specific architecture and can be paired with latent or pixel-space diffusion models. For our experiments we have selected Stable Diffusion 1.5 [45] and Stable Diffusion XL [46] (Dreamshaper-8 [47] and RealVisXL-V4 [48]) with the DDIM scheduler [43].

**Test set**   The experiments are conducted on images generated from the first 1000 prompts taken from the ContraStyles dataset [49]. Our color references are 1000 photos from Unsplash Lite [50]. We refer to these prompts and photos as the test set. A training set is not needed for our algorithm.

**Metrics**   To measure stylization strength, we calculate the Wasserstein-2 distance between color distributions in RGB space. Two content-related metrics are based on CLIP embeddings [42]. CLIP-IQA [51] is a cosine similarity between a generated image and pre-selected anchor vectors that define "good-looking" pictures. CLIP-T [42] is a cosine similarity between CLIP representations of a text prompt and an image generated from this prompt. In other words, the CLIP-T score indicates whether a modified sampling process still follows the initial text prompt, while CLIP-IQA measures the overall quality of the pictures.

**Baselines**   As discussed earlier, the problem of color-conditional generation can be solved by first creating an image from a text prompt and then performing a color transfer with a specialized color

transfer algorithm. Therefore, the largest family of baselines consists of algorithms of this kind applied to the output of SD1.5 and SDXL: Histogram matching (hm) [53], CT [56], MKL [55, 57], WCT2 [18], PhotoNAS [19], PhotoWCT2 [20], ModFlows [54]. The baseline "hm-mkl-hm" is a combination of histogram matching and MKL taken from the library [52]. In addition, we take three of the currently available baselines for stylized generation: IP-Adapter [30], InstantStyle [34], and RB-Modulation [39], though stylized generation is not exactly the problem we aim to solve. While recent work [58] also proposes an algorithm for color conditional generation with diffusion models, we exclude direct comparisons due to absence of open-source implementation. The term *Unconditional* indicates that no post-processing steps or controls were applied. We provide CLIP-IQA and CLIP-T metrics for *Unconditional SDXL* and *Unconditional SD1.5* as a reference.

**Comparison Results** The results in Table 1 prove that SW-Guidance achieves superior performance in color-conditional generation compared to all baseline methods. In particular, SW-Guidance has the minimal Wasserstein distance to the reference palette. At the same time, SW-Guidance has the highest CLIP-T among other algorithms for stylized generation (i.e. IP-Adapter, InstantStyle and RB-Modulation). This indicates the ability of SW-Guidance to follow the prompt without adding irrelevant features from the reference image in contrast to other stylization methods. In terms of overall image quality SW-Guidance holds the second place according to CLIP-IQA. Qualitative comparison with stylized generation is given in Fig. 3, with additional visual examples available in the Appendix. The Appendix also contains SD-1.5 performance scores and examples comparable to those shown in Table 1.

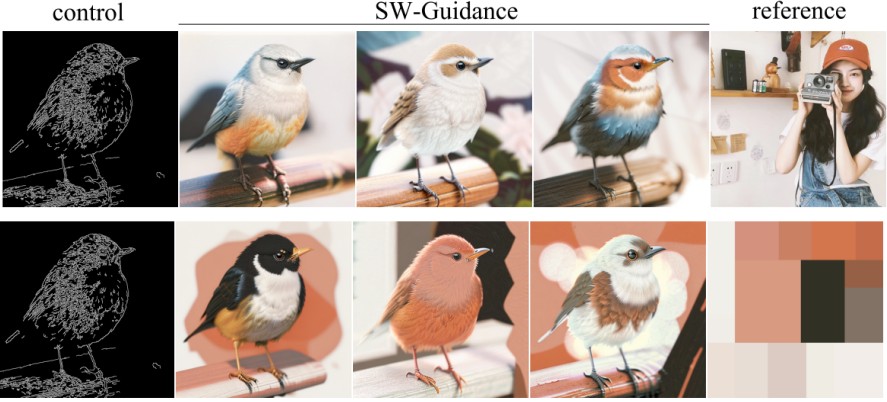

*SW-Guidance reference could be just a palette of colors*

Figure 4: SW-Guidance combined with canny ControlNet, SD1.5.

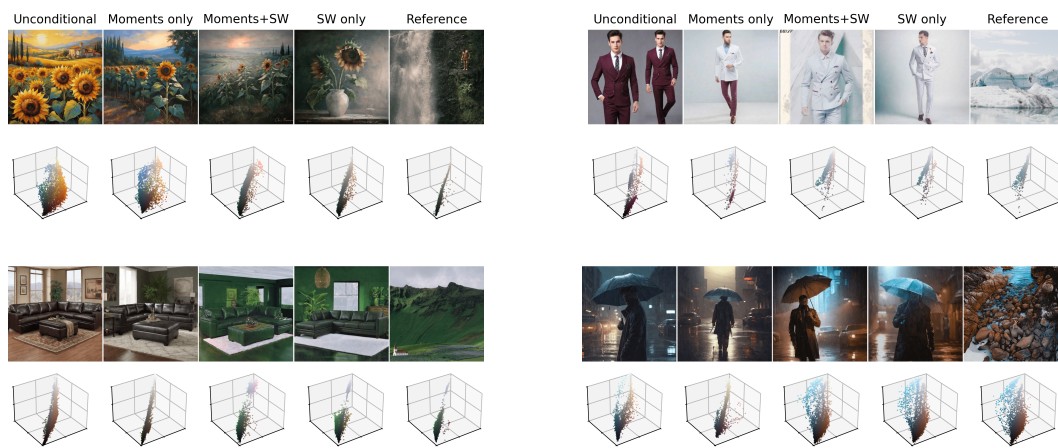

Figure 5: Ablation studies, SDXL. The best results are obtained with the loss function by Eq.7 (SW only). Moments-only guidance is insufficient. Please refer to Table 3 for the quantitative comparison.

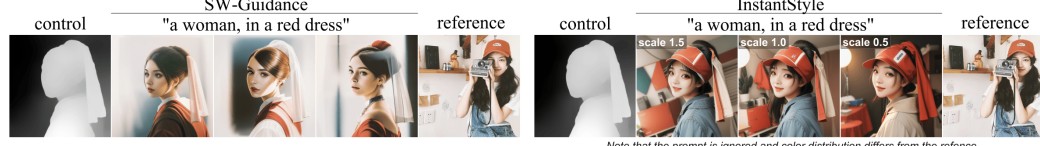

Figure 6: SW-Guidance combined with depth control has more flexibility than InstantStyle. SD1.5 model, scale is InstantStyle strength.

**Compatibility with ControlNets** SW-Guidance can be combined with other control methods to define the image layout, see Fig. 4 for the canny control and Fig. 6 for the depth map control. Our method supports any picture, representing a palette, as in the second row, Fig. 4.

Note that stylizing algorithms, such as InstantStyle, transfer not only color but also other features (see Fig. 3), making it difficult to control color separately. Fig. 6 shows that for InstantStyle the text prompt guiding the color is ignored because it contradicts the features of the reference image (i.e denim dress). Our method is more flexible and sets a red shade which aligns with the reference palette.

Relying only on text prompts for color control is inconvenient. Moreover, color naming is often connotative, and words like "lavender", "emerald" and "lime" can introduce unintended content details, as shown in Fig. 8. Please refer to the Appendix for more examples.

With all this said, we conclude that the proposed SW-Guidance is superior in color stylization while maintaining both integrity with the textual prompt and the quality of the produced images.

## 4.1 Ablation study

Table 2: Ablation study. SD-1.5. Analysis of different Sliced Wasserstein distances.

| 2-Wasserstein distance [4] ↓ | | Content scores | |
| --- | --- | --- | --- |
| Distance | mean $\pm$ std of mean | CLIP-IQA [51] ↑ | CLIP-T [42] ↑ |
| Sliced Wasserstein [5] | **0.0385 $\pm$ 0.0006** | 0.2220 $\pm$ 0.0027 | 0.2520 $\pm$ 0.0017 |
| Energy-Based SW [11] | 0.0390 $\pm$ 0.0006 | 0.2241 $\pm$ 0.0030 | 0.2535 $\pm$ 0.0017 |
| Distributional SW [9] | 0.0547 $\pm$ 0.0006 | 0.2225 $\pm$ 0.0030 | 0.2564 $\pm$ 0.0016 |
| Generalized SW [8] | 0.0879 $\pm$ 0.0014 | 0.2098 $\pm$ 0.0027 | **0.2594 $\pm$ 0.0016** |
| Mean & Cov | 0.1064 $\pm$ 0.0013 | **0.2258 $\pm$ 0.0030** | 0.2545 $\pm$ 0.0017 |

Table 3: Ablation study. SDXL. The impact of adding the first two moments to the SW distance (Eq.8), which is also shown in Fig. 5

| 2-Wasserstein distance [4] ↓ | | Content scores | |
| --- | --- | --- | --- |
| Distance | mean $\pm$ std of mean | CLIP-IQA [51] ↑ | CLIP-T [42] ↑ |
| SW only | **0.0297 $\pm$ 0.0005** | **0.285 $\pm$ 0.004** | 0.270 $\pm$ 0.002 |
| Moments + SW | 0.0305 $\pm$ 0.0006 | 0.279 $\pm$ 0.003 | 0.269 $\pm$ 0.002 |
| Moments only | 0.1176 $\pm$ 0.0016 | 0.276 $\pm$ 0.003 | 0.282 $\pm$ 0.002 |
| Unconditional SDXL [48] | 0.3824 $\pm$ 0.0056 | 0.239 $\pm$ 0.003 | **0.294 $\pm$ 0.002** |

**Different Sliced Wasserstein distances** Table 2 contains scores for the tested variants of Sliced Wasserstein (SW), each assessed under $K = 10$ slices, $M = 10$ iterations per scheduler step, and $lr = 100$ learning rate. Let us note that Lemma 2 holds for all of them. Please find their formal definition in Appendix section. In general, we didn't observe any substantial difference in their content scores. We can also note that, despite the time metric is absent in the table, Distributional Sliced Wasserstein (DSW) takes more time due to inner optimization loop. This suggests that although DSW and Generalized SW are aimed to converge faster for multidimensional distributions, this advantage does not translate to our 3D color transfer task. The Energy-Based SW [11] offered

a computationally light alternative, though it lacks any clear advantage over regular SW for this application.

**Generation time**   The generation time dependence on $M$ (inner steps) and $K$ (number of slices) for SD-1.5 is shown in Fig. 7. For our main experiments we set $M = 10$ and $K = 10$, which results in 30 seconds for SD-1.5 and around 1 minute for SDXL to generate an image on Nvidia RTX 4090 GPU. This represents an improvement compared to the 2 minutes required by RB-modulation.

**Mean and covariance terms**   The impact of adding the first two moments to the SW distance is presented in Fig. 5 and Table 3. The best results are obtained with the loss function by Eq.7 (SW only). Mean and covariance terms (Eq.8, Moments + SW) do not increase color similarity and tend to produce images of worse quality. Moments-only guidance is insufficient.

**Dependence on learning rate**   This experiment can be found in Appendix section.

## 5   Limitations and Discussion

The first important limitation of the proposed guidance is its sensitivity to the information about colors in text prompts, especially when they contradict the selected style reference. A clash between the textual and SW guidance typically results in visual artifacts, so detailed textual palette descriptions should be avoided.

Secondly, combining this method with existing stylizing attention-based approaches is not guaranteed to work, as strong stylizing methods could also lead to a clash of color guidance. Ideally, other conditioning should be disentangled from the color information. This collision effect is a subject for further research. As an example, we provide a joint run of InstantStyle and SW-Guidance (Fig. 9).

The last point we would like to discuss is the current implementation's requirement to differentiate through a U-net. Theoretically, this requirement could be avoided, but like the previous point, it requires additional study.

To sum up, this paper presents SW-Guidance, a novel training-free technique for color-conditional generation that can be applied to a range of denoising diffusion probabilistic models. Our study covers the SD-1.5 and SDXL architectures, and for both implementations, we achieved superior results in color similarity compared to color transfer algorithms and models for stylized generation. Numerically, we show the ability of SW-Guidance to maintain integrity with the textual prompt and preserve the quality of the produced images. Our qualitative examples demonstrate the absence of unwanted textures and irrelevant features from the reference image.

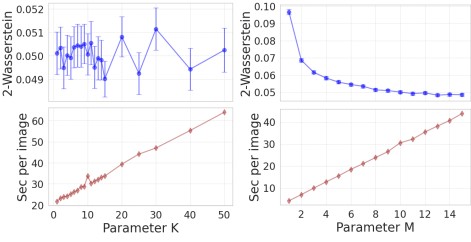

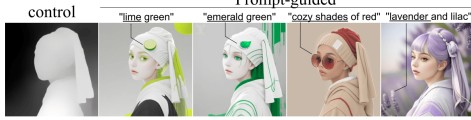

Figure 8: Text description of a color may introduce unwanted content details.

Figure 7: Ablation study for the dependence on $M$ (inner steps) and $K$ (number of slices) for SD-1.5. We use $M = 10$ and $K = 10$, which results in 30 seconds for SD-1.5 and around 1 minute for SDXL to generate an image on RTX 4090 GPU.

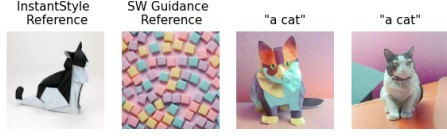

Figure 9: Limitations. Combination of SW-Guidance and InstantStyle SDXL.

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

# A  Sliced Wasserstein Distances

**Sliced Wasserstein Distance**  Wasserstein distances appear to be natural for our task of color transfer as they measure the cost of transporting one probability distribution to match another [4]. The Wasserstein distance of order $p$ is

$$W_p(\pi_0, \pi_1) = \left( \inf_{\pi \in \Pi(\pi_0, \pi_1)} \int_{\mathcal{X}_0 \times \mathcal{X}_1} ||x - y||^p \, d\pi(x, y) \right)^{1/p}, \tag{14}$$

where $\Pi(\pi_0, \pi_1)$ represents the set of all joint distributions with marginals $\pi_0$ and $\pi_1$. However, directly computing $W_p(\pi_0, \pi_1)$ is computationally expensive and difficult to differentiate through, because its value is itself a result of an optimization procedure $\inf$ over all transport plans $\Pi(\pi_0, \pi_1)$.

To overcome this issue, the sliced Wasserstein (SW) distance was introduced [5], offering a more computationally tractable alternative by reducing high-dimensional distributions to one-dimensional projections where the Wasserstein distance can be computed more straightforwardly. The sliced $p$-Wasserstein distance is defined as [5, 6]:

$$SW_p(\pi_0, \pi_1) = \left( \int_{\mathbb{S}^{d-1}} W_p^p(P_\theta \pi_0, P_\theta \pi_1) \, d\theta \right)^{1/p}, \tag{15}$$

where $\mathbb{S}^{d-1}$ is the unit sphere in $\mathbb{R}^d$ with $\int_{\mathbb{S}^{d-1}} d\theta = 1$, $P_\theta$ is a linear projection onto a one-dimensional subspace defined by $\theta$ ( Radon transformation in general) and $W_p^p$ is an ordinary p-Wasserstein distance by Eq.14.

A known issue with the Sliced Wasserstein (SW) distance arises when sampling parameters $\theta$ for projections. As noted in [8], uniformly sampled $\theta$ values on the unit sphere $\mathbb{S}^{d-1}$ in high dimensions tend to be nearly orthogonal. This resulting in $W_2(P_\theta \pi_0, P_\theta \pi_1) \approx 0$ with high probability. Consequently, these projections fail to provide discriminative information about the differences between the distributions $\pi_0$ and $\pi_1$.

**Distributional Sliced Wasserstein Distance**  The Distributional Sliced Wasserstein (DSW) distance, proposed in [9] generalizes the SW distance by introducing a probability distribution $\sigma(\theta)$ over the slicing directions and defined as:

$$DSW_p(\pi_0, \pi_1) =$$
$$= \sup_\sigma \left( \int_{\mathbb{S}^{d-1}} W_p^p(P_\theta \pi_0, P_\theta \pi_1) \, \sigma(\theta) d\theta \right)^{1/p}, \tag{16}$$

where the optimization $\sup$ is performed w.r.t probability distributions $\sigma$ over unit sphere $\mathbb{S}^{d-1}$, with $\int_{\mathbb{S}^{d-1}} \sigma(\theta) d\theta = 1$.

**Energy-Based Sliced Wasserstein Distance**

The Energy-Based Sliced Wasserstein (EBSW) distance, introduced in [11], provides an alternative to the optimization-based approach of DSW by defining a slicing distribution $\sigma_{\pi_0, \pi_1}(\theta; f, p)$ based on the projected Wasserstein distances:

$$\sigma_{\pi_0, \pi_1}(\theta; f, p) \propto f(W_p^p(P_\theta \pi_0, P_\theta \pi_1)), \tag{17}$$

where $f$ is a monotonically increasing energy function (e.g., $f(x) = e^x$) that emphasizes directions with larger projected Wasserstein distances. Using this slicing distribution, the EBSW distance is defined as:

$$EBSW_p(\pi_0, \pi_1; f) =$$
$$\mathbb{E}_{\theta \sim \sigma_{\pi_0, \pi_1}(\theta; f, p)} \left[ W_p^p(P_\theta \pi_0, P_\theta \pi_1) \right]^{1/p}. \tag{18}$$

To improve computational efficiency, importance sampling is used, with a proposal distribution $\sigma_0(\theta)$ to sample directions and weight them according to the ratio:

$$w_{\pi_0,\pi_1,\sigma_0,f,p}(\theta) = \frac{f(W_p^p(P_\theta\pi_0, P_\theta\pi_1))}{\sigma_0(\theta)}. \tag{19}$$

**Generalized Sliced Wasserstein Distance**   The Generalized Sliced Wasserstein (GSW) distance [8] replaces the Radon transform with a generalized Radon transform that depends on a defining function $g(x,\theta)$. Formally, for a function $I$, the generalized Radon transform is defined as:

$$GI(t,\theta) = \int_{\mathbb{R}^d} I(x)\delta(t - g(x,\theta))\,dx, \tag{20}$$

where $\delta$ is the Dirac delta function. Using the generalized Radon transform, the GSW distance between two distributions $\pi_0$ and $\pi_1$ is defined as:

$$GSW_p(\pi_0, \pi_1) = \left( \int_{\Omega_\theta} W_p^p(GI_{\pi_0}(\cdot,\theta), GI_{\pi_1}(\cdot,\theta))\,d\theta \right)^{1/p}, \tag{21}$$

where $\Omega_\theta$ is a compact set of feasible parameters for the function $g(x,\theta)$ (e.g., $\Omega_\theta = \mathbb{S}^{d-1}$ for $g(x,\theta) = \langle x,\theta \rangle$).

For empirical distributions $\pi_0$ and $\pi_1$, represented by samples $\{x_i\}_{i=1}^N$ and $\{y_j\}_{j=1}^N$, the GSW distance can be approximated as:

$$GSW_p(\pi_0, \pi_1) \approx$$
$$\left( \frac{1}{L} \sum_{l=1}^{L} \sum_{n=1}^{N} \left| g(x_{i[n]}, \theta_l) - g(y_{j[n]}, \theta_l) \right|^p \right)^{1/p}, \tag{22}$$

where $x_{i[n]}$ and $y_{j[n]}$ denote the sorted indices of $\{g(x_i,\theta_l)\}_{i=1}^N$ and $\{g(y_j,\theta_l)\}_{j=1}^N$, respectively, for each sampled $\theta_l$.

## B   Theoretical Justification

This section contains proofs of Proposition 1 and Lemma 2 from the main text (here they are numbered as Proposition 4 and Lemma 5). Though the statement of Proposition 4 can be found in the literature, its formal treatment is omitted [4, 59]. Here we provide its detailed proof for Borel probability measures on $\mathbb{R}$. It restricts us to non-decreasing, right-continuous cumulative distribution functions $F$, Fig 10.

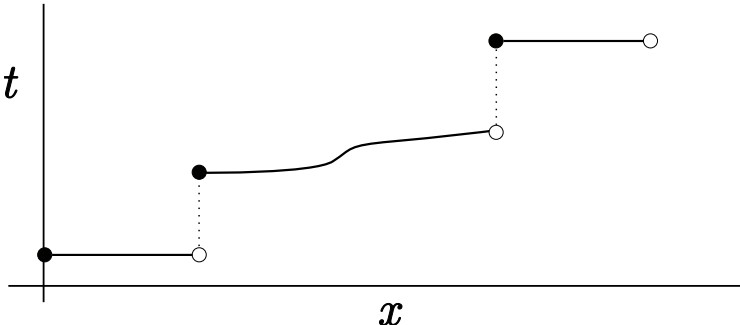

Figure 10: Example of right continuous non-decreasing function.

We need Proposition 4 for efficient sampling, as it allows one to avoid computing the inverse CDF. First we prove Lemmas 1, 2 and 3.

**Lemma 1.** *Let $F$ be a cumulative distribution function (CDF) on $\mathbb{R}$, and let $F^{-1}(t) = \inf\{x \in \mathbb{R} \mid F(x) \geq t\}$ be its quantile function for $t \in [0, 1]$. Then:*

$$\{t \in [0, 1] \mid F^{-1}(t) \leq a\} = \{t \in [0, 1] \mid F(a) \geq t\}. \tag{23}$$

*Proof.* L.H.S. $\Rightarrow$ R.H.S.:

Suppose $t' \in \{t \in [0, 1] \mid F^{-1}(t) \leq a\}$. Then, there exists $x' = F^{-1}(t')$ such that $x' \leq a$. By the definition of the quantile function $F^{-1}(t')$, $x'$ is the infimum of the set $\{x \mid t' \leq F(x)\}$. Under the assumptions that $F$ is right-continuous, the infimum $x'$ belongs to the set, and therefore $F(x') \geq t'$. Since $F(x)$ is non-decreasing and $a \geq x'$, it follows that $F(a) \geq F(x') \geq t'$. Hence, $t' \in \{t \in [0, 1] \mid F(a) \geq t\}$.

R.H.S. $\Rightarrow$ L.H.S.:

Suppose $t' \in \{t \in [0, 1] \mid F(a) \geq t\}$, but $t' \notin \{t \in [0, 1] \mid F^{-1}(t) \leq a\}$, i.e. $t'$ such that $t' \leq F(a)$ and $F^{-1}(t') > a$. However, by the definition of $x' = F^{-1}(\hat{t})$, $x'$ is the infimum of the set $\{x \mid F(x) \geq t'\}$. Since $a < x'$, $a$ cannot belong to this set, implying $F(a) < t'$, which contradicts the assumption $t' \leq F(a)$. Thus, there is no $t$ in the R.H.S. that does not also belong to the L.H.S.

From these, we conclude that the two sets are equal:

$$\{t \in [0, 1] \mid F^{-1}(t) \leq a\} = \{t \in [0, 1] \mid F(a) \geq t\}. \tag{24}$$

$\square$

**Lemma 2.** *Let $F$ be a cumulative distribution function (CDF) on $\mathbb{R}$. Then the quantile function $F^{-1}(t) = \inf\{x \in \mathbb{R} \mid F(x) \geq t\}$, defined for $t \in [0, 1]$, is measurable with respect to the Borel sigma algebra.*

*Proof.* To show that $F^{-1}(t)\colon ([0, 1], \mathcal{B}_{[0,1]}) \to (\mathbb{R}, \mathcal{B}_{\mathbb{R}})$ is measurable, we must prove that for any Borel set $B \subset \mathbb{R}$, the preimage:

$$\{t \in [0, 1] \mid F^{-1}(t) \in B\} \in \mathcal{B}_{[0,1]}. \tag{25}$$

The Borel sigma algebra $\mathcal{B}_{\mathbb{R}}$ is generated by intervals of the form $(-\infty, b]$. Hence, it suffices to prove that for any $b \in \mathbb{R}$, the set

$$\{t \in [0, 1] \mid F^{-1}(t) \in (-\infty, b]\} \tag{26}$$

is measurable.

Consider the preimages of $(-\infty, b]$:

$$
\begin{aligned}
\{t \in [0, 1] \mid F^{-1}(t) \in (-\infty, b]\} &= \\
&= \{t \in [0, 1] \mid F^{-1}(t) \leq b\} = \text{/by Lemma 1/} \\
&= \{t \in [0, 1] \mid F(b) \geq t\} = [0, F(b)]
\end{aligned}
\tag{27}
$$

Since $F$ is a CDF, $F(b)$ is a real number in $[0, 1]$, and the set $\{t \in [0, 1] \mid t \leq F(b)\} = [0, F(b)]$ is a Borel set in $[0, 1]$, and thus the preimage of $(-\infty, b]$ is a measurable set in $[0, 1]$. Therefore the quantile function $F^{-1}(t)$ is measurable. $\square$

**Lemma 3.** *Let $a$ and $b$ be two real numbers. Then:*

$$|a - b| = \int_{\mathbb{R}} |I_{a \geq u} - I_{b \geq u}| \, du, \tag{28}$$

*where $I_{x \geq u}$ is the indicator of the set $\{x \in \mathbb{R} \mid x \geq u\}$.*

*Proof.* First, suppose $a \geq b$. Consider three cases for $u$:

1. If $u > a > b$, then $I_{a \geq u} = 0$ and $I_{b \geq u} = 0$, so
   $|I_{a \geq u} - I_{b \geq u}| = 0$.

2. If $a > b > u$, then $I_{a \geq u} = 1$ and $I_{b \geq u} = 1$, so
$|I_{a \geq u} - I_{b \geq u}| = 0.$

3. If $a > u > b$, then $I_{a \geq u} = 1$ and $I_{b \geq u} = 0$, so
$|I_{a \geq u} - I_{b \geq u}| = 1.$

Therefore, the integral reduces to:

$$\int_{\mathbb{R}} |I_{a \geq u} - I_{b \geq u}| \, du = \int_b^a 1 \, du = a - b. \tag{29}$$

For the case $b > a$, by a similar argument, integral is not zero only when:

$$b \geq u \geq a \quad |I_{a \geq u} - I_{b \geq u}| = 1.$$

and therefore, the integral reduces to

$$\int_{\mathbb{R}} |I_{a \geq u} - I_{b \geq u}| \, du = \int_a^b 1 \, du = b - a. \tag{30}$$

Thus, in all cases:

$$|a - b| = \int_{\mathbb{R}} |I_{a \geq u} - I_{b \geq u}| \, du. \tag{31}$$

$\square$

**Proposition 4.** *Let $F$ and $G$ be cumulative distribution functions (CDFs) on $\mathbb{R}$. Then:*

$$\int_0^1 |F^{-1}(t) - G^{-1}(t)| \, dt = \int_{\mathbb{R}} |F(x) - G(x)| \, dx, \tag{32}$$

*where $F^{-1}$ and $G^{-1}$ are the quantile functions (generalized inverse CDFs) of $F$ and $G$, respectively.*

*Proof.* Note, that by Lemma 2 both $F^{-1}$ and $G^{-1}$ are measurable and therefore the L.H.S exists. By Lemma 3, its absolute value can be represented as:

$$\int_0^1 |F^{-1}(t) - G^{-1}(t)| \, dt =$$
$$= \int_0^1 \int_{\mathbb{R}} |I_{F^{-1}(t) \geq u} - I_{G^{-1}(t) \geq u}| \, du \, dt. \tag{33}$$

Using the property of indicator functions $I_{F^{-1}(t) \geq u} = 1 - I_{F^{-1}(t) < u}$, the integral becomes:

$$\int_0^1 \int_{\mathbb{R}} |I_{F^{-1}(t) \geq u} - I_{G^{-1}(t) \geq u}| \, du \, dt$$
$$= \int_0^1 \int_{\mathbb{R}} |-I_{F^{-1}(t) < u} + I_{G^{-1}(t) < u}| \, du \, dt \tag{34}$$
$$= \int_0^1 \int_{\mathbb{R}} |-I_{F^{-1}(t) \leq u} + I_{G^{-1}(t) \leq u}| \, du \, dt.$$

where the last equality is correct since function under the Lebesgue integral can be changed on a set of measure zero. Using Lemma 1 we rewrite indicators:

$$\int_0^1 \int_{\mathbb{R}} |-I_{t \leq F(u)} + I_{t \leq G(u)}| \, du \, dt \tag{35}$$

By Fubini's theorem (justified as the integrand is non-negative and measurable), we can switch the order of integration:

$$\int_{\mathbb{R}} \int_0^1 |-I_{t \leq F(u)} + I_{t \leq G(u)}| \, dt \, du. \tag{36}$$

Using the Lemma 3 again we get:

$$\int_{\mathbb{R}} \int_0^1 \left| -I_{t \leq F(u)} + I_{t \leq G(u)} \right| \, dt \, du$$
$$= \int_{\mathbb{R}} |G(u) - F(u)| \, du. \tag{37}$$

Hence, we conclude:

$$\int_0^1 \left| F^{-1}(t) - G^{-1}(t) \right| \, dt = \int_{\mathbb{R}} |F(x) - G(x)| \, dx. \tag{38}$$

$\square$

Lemma 5 (Lemma 2 in the main text) provides the theoretical foundation for our optimization procedure for multidimensional Borel probability measures $\mu_n$ and $\mu$ on $\mathbb{R}^d$.

**Lemma 5.** *Let $\mu_n$ and $\mu$ be Borel probability measures on the unit cube $[0,1]^d \subset \mathbb{R}^d$. If*

$$\lim_{n \to \infty} \mathrm{SW}(\mu_n, \mu) = 0, \tag{39}$$

*then $\mu_n$ converges weakly to $\mu$, and all moments of $\mu_n$ converge to the moments of $\mu$.*

*Proof.* Consider the ball $B(0, R)$ of radius $R$, that contains the unit cube. Then a Borel probability measure on the cube $[0,1]^d$ can be extended to the Borel probability measure on $B(0, R)$ by assigning measure zero to any Borel set outside of the cube.

Now we can use Lemma 5.1.4 from [60], which states that for the 1-Wasserstein distance $W_1$ there exists a constant $C_d > 0$ such that for all Borel probability measures $\mu, \nu$ on $B(0, R)$

$$0 \leq \mathrm{W}_1(\mu, \nu) \leq C_d \, R^{\frac{d}{d+1}} \, \mathrm{SW}_1(\mu, \nu)^{\frac{1}{d+1}}. \tag{40}$$

Since $\mu_n$ and $\mu$ are supported on the unit cube in $\mathbb{R}^d$, we take $R = \sqrt{d}$, which is a sufficient radius to bound the unit cube. From the assumption that $\lim_{n \to \infty} \mathrm{SW}_1(\mu_n, \mu) = 0$, we have:

$$\lim_{n \to \infty} C_d \, R^{\frac{d}{d+1}} \, \mathrm{SW}_1(\mu_n, \mu)^{\frac{1}{d+1}} = 0. \tag{41}$$

Using the squeeze Theorem for (40), it follows that:

$$\lim_{n \to \infty} \mathrm{W}_1(\mu_n, \mu) = 0. \tag{42}$$

By Definition 6.8 (iv) and Theorem 6.9 of [4], the convergence $\mathrm{W}_1(\mu_n, \mu) \to 0$ implies that $\mu_n$ converges weakly to $\mu$. Specifically, for any $x_0 \in B(0, R)$ and all continuous functions $\varphi$ with $|\varphi| \leq C \, (1 + d(x_0, x))$, $C \in \mathbb{R}$ one has

$$\lim_{n \to \infty} \int \varphi(x) \, d\mu_n(x) = \int \varphi(x) \, d\mu(x). \tag{43}$$

For our case $d(x_0, x) \leq 2R$, so $\varphi$ is bounded, and integration over the $B(0, R)$ could be replaced with integration over the unit cube by a construction of our extension of $\mu_n$ and $\mu$.

Given a (finite) multi-index $\bar{\alpha} = (\alpha_1, \alpha_2, \ldots, \alpha_d)$, one can define the moment:

$$m_{\bar{\alpha}} = \int x_1^{\alpha_1} x_2^{\alpha_2} \cdots x_d^{\alpha_d} \, d\mu(x). \tag{44}$$

Polynomial functions $\phi(x) = x^{\bar{\alpha}}$ are bounded and continuous on the unit cube because $x_i \leq 1$ for all $i \in \{1, \ldots, d\}$, ensuring all terms $x^{\bar{\alpha}} \leq 1$. Thus, weak convergence implies that for all multi-indices $\bar{\alpha}$,

$$\lim_{n \to \infty} \int x^{\bar{\alpha}} \, d\mu_n(x) = \int x^{\bar{\alpha}} \, d\mu(x), \tag{45}$$

i.e., all moments of $\mu_n$ converge to the corresponding moments of $\mu$ component-wise. $\square$

**Algorithm 2** Color Conditional Generation with Sliced Wasserstein Guidance for latent text-to-image diffusion

**Require:**
    DDIM: Diffusion DDIM scheduler
    $s_\theta$: UNet model
    $D$: Decoder of the Variational Autoencoder
    $E$: Encoder of the Variational Autoencoder
    $\tau$: Text embeddings for conditioning
    $I_{\text{ref}}$: Reference image
    $\gamma$: Guidance scale factor
    $M$: Number of optimization steps
    Initialize $x_t \sim \mathcal{N}(0, I)$

1: **for** $t$ **in** $\{0, \dots, T-1\}$ **do**
2:     $u \leftarrow \mathbf{0}$ (tensor with same shape as $x_t$)
3:     **for** $j$ **in** $\{1, \dots, M\}$ **do**
4:         $x_t' \leftarrow x_t + u$
5:         $\epsilon \leftarrow s_\theta(x_t', t, \tau)$
6:         $\hat{x}_0 \leftarrow \text{DDIM}(\epsilon, t, x_t')$
7:         $I_{\text{gen}} \leftarrow D(\hat{x}_0)$
8:         $P_{\text{gen}} \leftarrow \text{pixels\_from\_image}(I_{\text{gen}})$
9:         $K \leftarrow 10$                                       ▷ Number of slices
10:         **for** $k$ **in** $\{1, \dots, K\}$ **do**
11:             $R \leftarrow \text{rand\_rotation\_matrix}()$
12:             $P_{\text{gen}}^R \leftarrow P_{\text{gen}}^T R$
13:             $P_{\text{ref}}^R \leftarrow P_{\text{ref}}^T R$
14:             **for** $d$ **in** $\{1, \dots, 3\}$ **do**
15:                 $x_{\text{rot}} \leftarrow P_{\text{gen}}^R[:, d]$
16:                 $y_{\text{rot}} \leftarrow P_{\text{ref}}^R[:, d]$
17:                 $\text{cdf}_x \leftarrow \text{get\_cdf}(x_{\text{rot}})$
18:                 $\text{cdf}_y \leftarrow \text{get\_cdf}(y_{\text{rot}})$
19:                 $\mathcal{L} \leftarrow \mathcal{L} + \text{mean}(|\text{cdf}_x - \text{cdf}_y|)$
20:             **end for**
21:         **end for**
22:         $g_u \leftarrow \nabla_u \mathcal{L}(u)$
23:         $g_u \leftarrow \frac{g_u}{\text{std}(g_u)}$
24:         $u \leftarrow u - \lambda g_u$
25:     **end for**
26:     $x_t^* \leftarrow x_t + u$
27:     $\epsilon_{\text{cond}} \leftarrow s_\theta(x_t^*, t, \tau)$
28:     $\epsilon_{\text{uncond}} \leftarrow s_\theta(x_t^*, t, \emptyset)$
29:     $\epsilon_{\text{guided}} \leftarrow \epsilon_{\text{uncond}} + \gamma(\epsilon_{\text{cond}} - \epsilon_{\text{uncond}})$
30:     $x_t \leftarrow \text{DDIM}(\epsilon_{\text{guided}}, t, x_t^*)$
31: **end for**

Table 4: Text-to-image generation conditioned on a reference color distribution. Quantitative evaluation, SD1.5 [47]. 2-Wasserstein distance between the color distributions measures color similarity, CLIP-IQA and CLIP-T are quality and content scores. All color transfer methods [18, 19, 20, 52, 53, 54, 55, 56] are applied to the Unconditional SD1.5 generations.

| | 2-Wasserstein distance [4] ↓ | Content scores | |
|---|---|---|---|
| Algorithm | mean ± std of mean | CLIP-IQA [51] ↑ | CLIP-T [42] ↑ |
| SW-Guidance SD-1.5 (ours) | **0.0328 ± 0.0003** | 0.2221 ± 0.0029 | 0.2624 ± 0.0017 |
| hm-mkl-hm [52] | 0.0572 ± 0.0011 | 0.2013 ± 0.0030 | 0.2656 ± 0.0017 |
| hm [53] | 0.0896 ± 0.0019 | 0.2054 ± 0.0029 | 0.2700 ± 0.0016 |
| PhotoWCT2 [20] | 0.1085 ± 0.0016 | 0.1796 ± 0.0026 | 0.2621 ± 0.0016 |
| ModFlows [54] | 0.1182 ± 0.0015 | 0.2035 ± 0.0030 | 0.2640 ± 0.0016 |
| Colorcanny | | | |
| ControlNet SD-1.5 [61] | 0.1183 ± 0.0016 | 0.1953 ± 0.0025 | 0.2600 ± 0.0018 |
| MKL [55] | 0.1274 ± 0.0018 | 0.1880 ± 0.0028 | 0.2700 ± 0.0016 |
| CT [56] | 0.1412 ± 0.0019 | 0.1826 ± 0.0027 | 0.2713 ± 0.0016 |
| WCT2 [18] | 0.1425 ± 0.0018 | 0.1819 ± 0.0026 | **0.2761 ± 0.0016** |
| PhotoNAS [19] | 0.1724 ± 0.0017 | **0.2878 ± 0.0027** | 0.2590 ± 0.0015 |
| InstantStyle SD-1.5 [34] | 0.2802 ± 0.0043 | 0.1891 ± 0.0020 | 0.2554 ± 0.0018 |
| Unconditional SD-1.5 | 0.4062 ± 0.0063 | 0.2010 ± 0.0023 | 0.2837 ± 0.0016 |

## C   Additional results

**Dependence on learning rate**   The effect of learning rates on the performance of sliced Wasserstein-based guidance is given in Fig. 16. The learning rate has a significant impact on the 2-Wasserstein distance, with an optimal value of 0.04, beyond which the loss plateaus and then increases. In contrast, the CLIP-IQA and CLIP-T metrics exhibit linear relationships with respect to the learning rate, suggesting no minimum or optimal value within the range tested.

**Text prompts to control the color**   Using text prompts for controlling the color has several major issues. The first row of Fig. 15 shows that the red color specified by the prompt is often ignored. The second row of Fig. 15 shows how the same prompt applied to another control image produces completely different color distribution. It also introduces content details due to connotative words like "denim", "warm" and "soft". Removing these words alters the colors, making the prompt design tedious. Please note, that color naming is often connotative, and words like "bloody red" and "lime" will introduce content details.

**Content Diversity Evaluation**   To evaluate the content diversity of the generated images, we computed the FID between unconditional SDXL generations and those obtained using various style guidance methods. To mitigate potential effects of color distribution alignment on the FID, all evaluations were conducted after conversion to grayscale histogram normalized images. The results on our generated dataset are summarized in Table 5.

Table 5: FID scores between unconditional SDXL generations and stylized outputs.

| Method Used with SDXL | FID Score (vs. Unconditional) |
|---|---|
| Mean/Covariance Matching Only | 53.16 |
| SW-Guidance (Ours) | 58.40 |
| InstantStyle | 58.95 |
| IP-Adapter | 71.06 |
| RB Modulation | 72.75 |

The results show that SW-Guidance maintains content diversity comparable to other state-of-the-art stylization methods. While there is a slight FID increase compared to simple moment matching (which provides weaker color control), our method preserves substantially more diversity than stronger stylization techniques such as IP-Adapter and RB Modulation.

## D Experimental Details

The experiments were conducted on images generated by SD-1.5 (Dreamshaper-8) and SDXL (RealVisXL-V4) using the first 1000 ContraStyles prompts [49]. No negative prompts or negative embeddings were used.

We fixed the CFG scale to 5 and the resolution to 768x768 for SDXL. For SD-1.5, the CFG scale was set to 8 and the resolution to 512x512. Both the SDXL and SD pipelines used the DDIM scheduler with 30 inference steps. Images for RB-Modulation were produced by Stable Cascade with a resolution of 1024x1024 and a total of 30 inference steps (20 for stage C and 10 for stage B). Method-specific settings are provided below.

**Baselines** For InstantStyle, the SDXL and SD-1.5 scales were set to 1.0. For IP-Adapter, the SDXL scale was set to 0.5 because higher scales tended to ignore the text prompt, producing variations of a reference image. The Colorcanny ControlNet for SD-1.5 had a conditioning scale of 1.0. For SW-Guidance, the SD-1.5 learning rate was $lr = 0.04$. In the SDXL version of SW-Guidance, we did not apply gradient normalization (line 23, Algorithm 2) and set the constant $lr = 0.01 \cdot 10^4 = 100$.

For evaluation, we used publicly available models and algorithms (i.e., none of them were re-trained or re-implemented). We ran color transfer baselines with the default settings provided by the authors.

We observed that PhotoNAS demonstrated a dependency on the resolution of input images. Specifically, the method was optimized for 512×512 inputs and exhibited noticeable variations in performance, including high-frequency defects when images of different resolutions were used. Therefore, the evaluations for SDXL and DreamShaper were different, as SDXL outputs images in higher resolutions.

**Metrics** The 2-Wasserstein distance was estimated with 3000 randomly sampled points using the "emd" function from the POT library [62]. The CLIP-IQA metric implementation was taken from the 'piq' Python library [63]. The CLIP-T metric was calculated using the model "openai/clip-vit-large-patch14" with an embedding dimension of 768.

**Hardware** The experiments were conducted on a single workstation equipped with two Nvidia RTX 4090 GPU accelerators and 256 GB of RAM.

**Prompts for illustrations** *Fig. 1, (main text)*:

1. Astronaut in a jungle, detailed, 8k
2. A cinematic shot of a cute little rabbit wearing a jacket and doing a thumbs up
3. extremely detailed illustration of a steampunk train at the station, intricate details, perfect environment

*Fig. 5, (main text)*:

1. Sunflower Paintings | Sunflowers Painting by Chris Mc Morrow - Tuscan Sunflowers Fine Art ...
2. b8547793944 Formal dress suit men male slim wedding suits for men double breasted mens suits wine red costume ternos masculino fashion 2XL
3. martino leather chaise sectional sofa 2 piece apartment and sets from china interio tucson dining room rustic furniture with home the company
4. 1125x2436 Rainy Night Man With Umbrella Scifi Drawings Digital Art

*Fig. 17, Fig.12 and Fig. 11 (Appendix)* :

1. Woman with a Parasol - Madame Monet and Her Son - Image: Monet woman with a parasol right
2. """Iceland: Through an Artist's Eyes part 4 Rainy Day Adventures"" original fine art by Karen Margulis"
3. Parthenon Poster featuring the digital art Parthenon Of Nashville by Honour Hall
4. How To Make A Caramel Frappuccino At Home

5. New York Central Building, Park Avenue, 1930,Vintage Poster, by Chesley Bonestell

*Fig. 13 (Appendix)* :

1. Francis Day - The Piano Lesson Frederick Childe Hassam - The Sonata George Bellows - Emma at the Piano Theodore Robinson - Girl At Piano Pierre-Auguste Renoir - The Piano Lesson - Louise Abbema - At the Piano Gustave...

2. Illustration pour Girl retro military pilot pop art retro style. The army and air force. A woman in the army - image libre de droit

3. Victor Tsvetkov The Bicycle Ride 1965 Russian Painting, Russian Art, Figure Painting, Bicycle Painting, Bicycle Art, Socialist Realism, Soviet Art, Illustration Art, Illustrations

4. """""""There Was A Time"""""" Milwaukee, Wisconsin Horizons by Phil Koch USA"""

5. Poster featuring the painting Monet Wedding by Clara Sue Beym

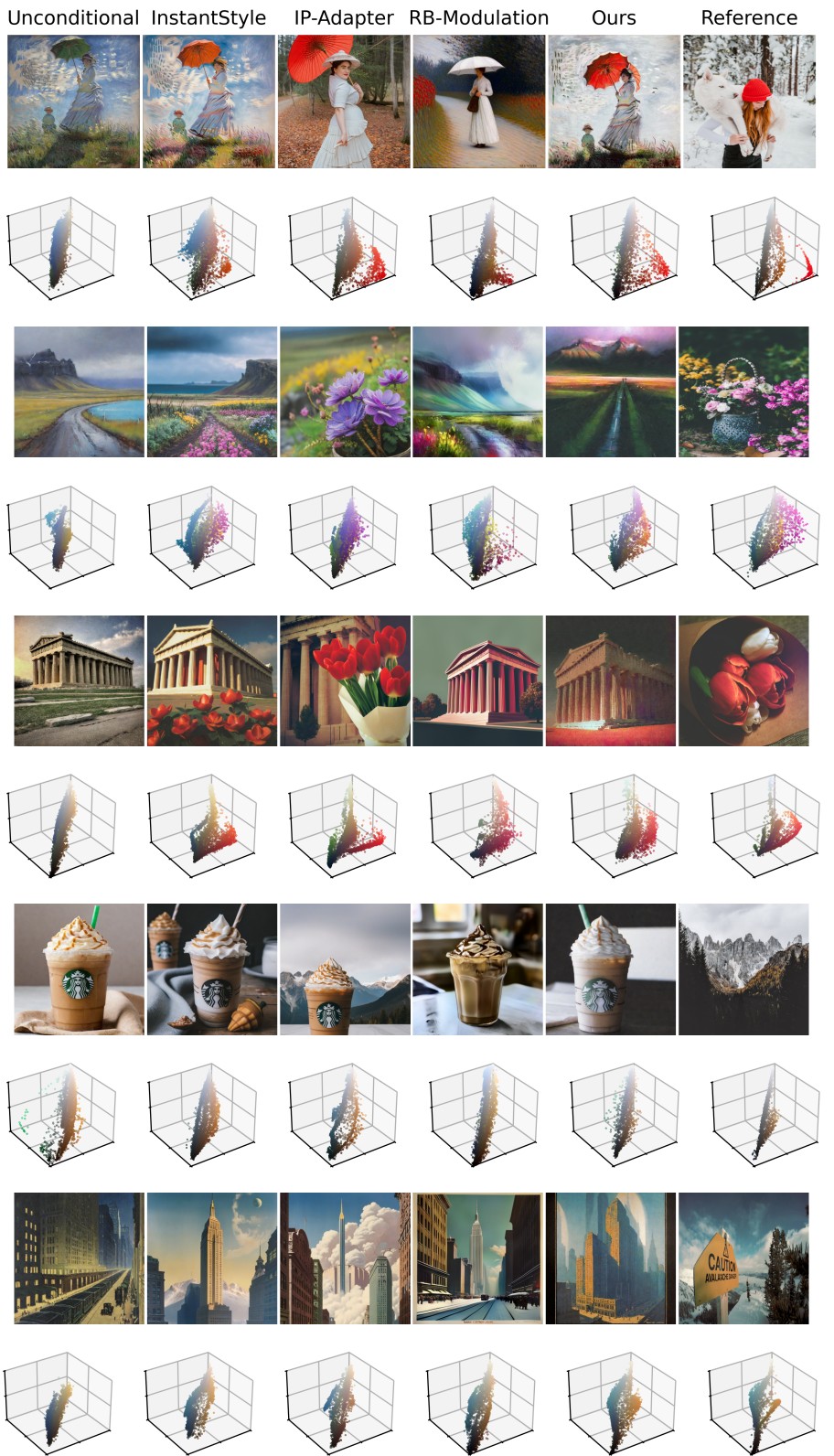

Figure 11: Text-to-image generation conditioned on a reference color distribution. Comparison with stylized generation methods. Examples from the test set. All images are generated by RealVisXL except of RB-Modulation running on Stable Cascade. Other methods have greater mismatch in color distributions and also often transfer some composition details such as: a forest (first row), a field of flowers (second row), a bouquet (third row), mountains (fourth row), cloudy sky and mountains (last row).

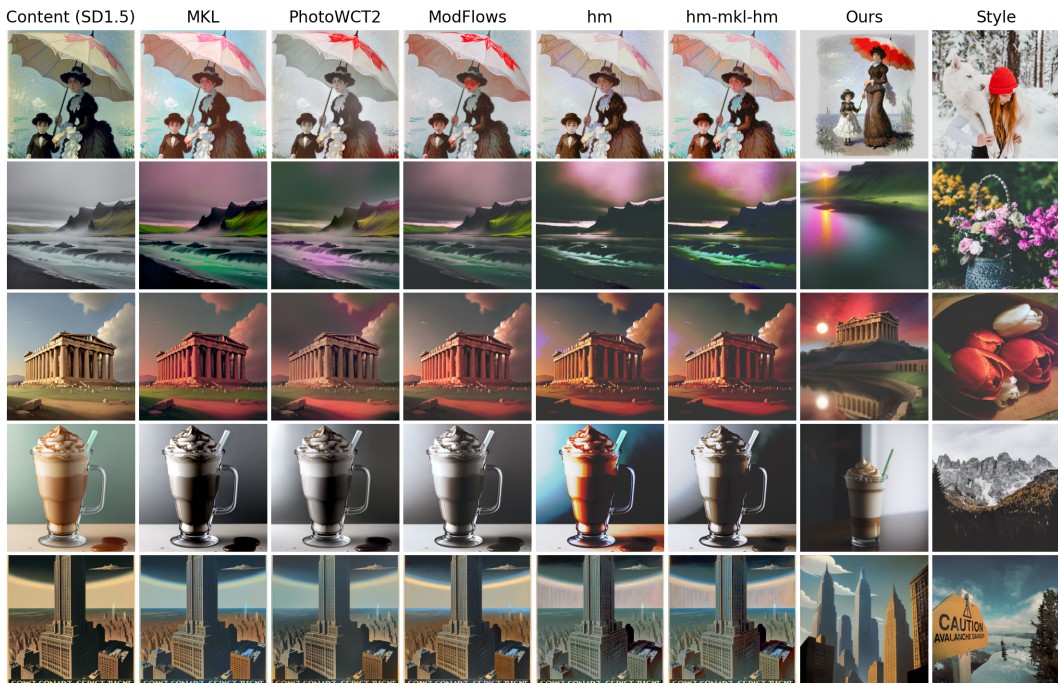

Figure 12: Text-to-image generation conditioned on a reference color distribution. Qualitative comparison with color transfer methods for SD-1.5. Examples from the test set. Please refer to the Table 4 for the quantitative comparison.

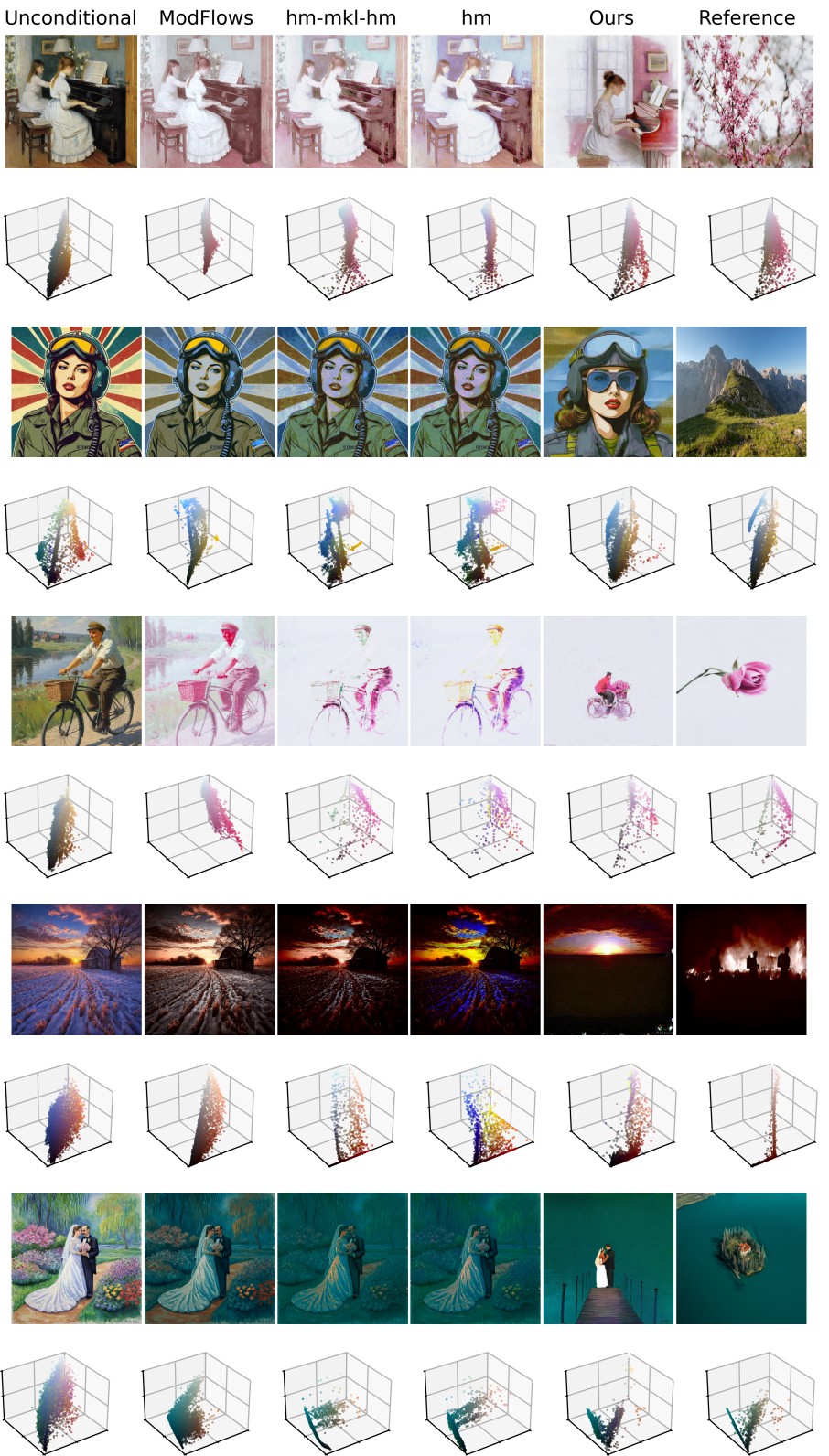

Figure 13: Text-to-image generation conditioned on a reference color distribution. Comparison with color transfer methods. Examples from the test set. Color transfer methods (ModFlows, hm and hm-mkl-hm) are applied to the Unconditional RealVisXL generations. Images produced by color transfer methods have greater mismatch in color distributions with the reference when compared to SW-Guidance.

control        SW-Guidance + ControlNet        reference

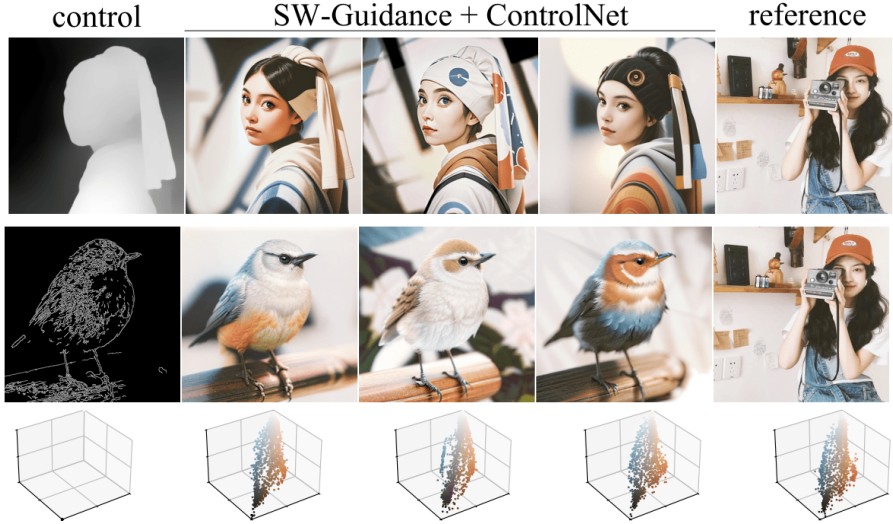

Figure 14: SW-Guidance combined with depth and canny controls.

control        Prompt + ControlNet        prompt

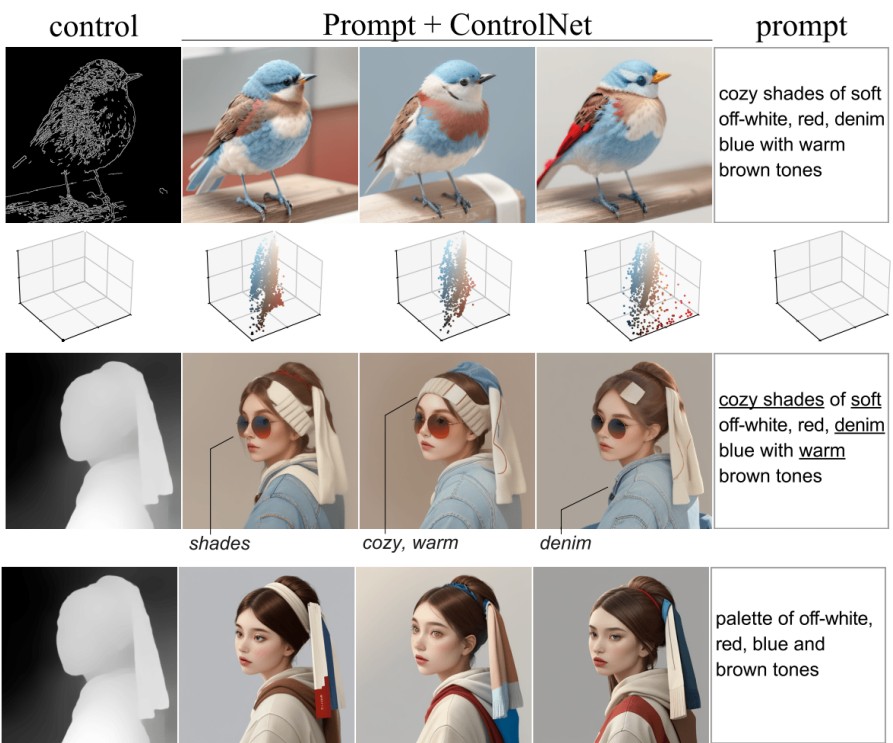

*Note: The absence of connotative terms like 'cozy,' 'denim,' and 'warm' alters the color distribution.*

Figure 15: Text prompt mimicking the color distribution of Fig. 14.

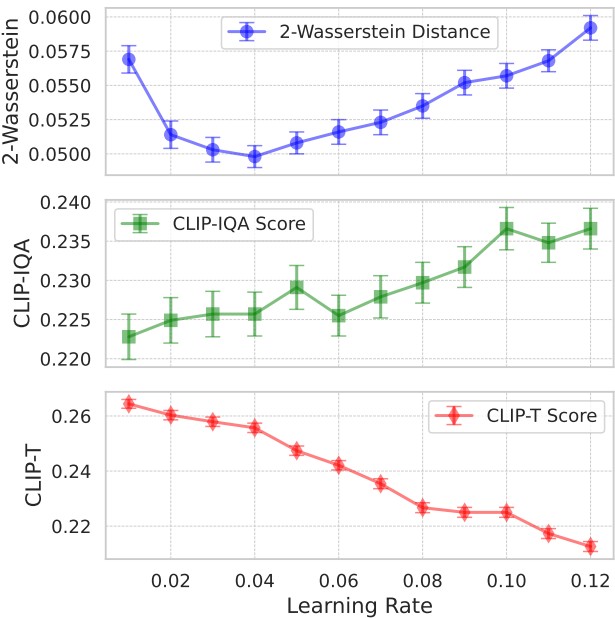

Figure 16: The performance metrics dependence on the learning rate for SD-1.5.

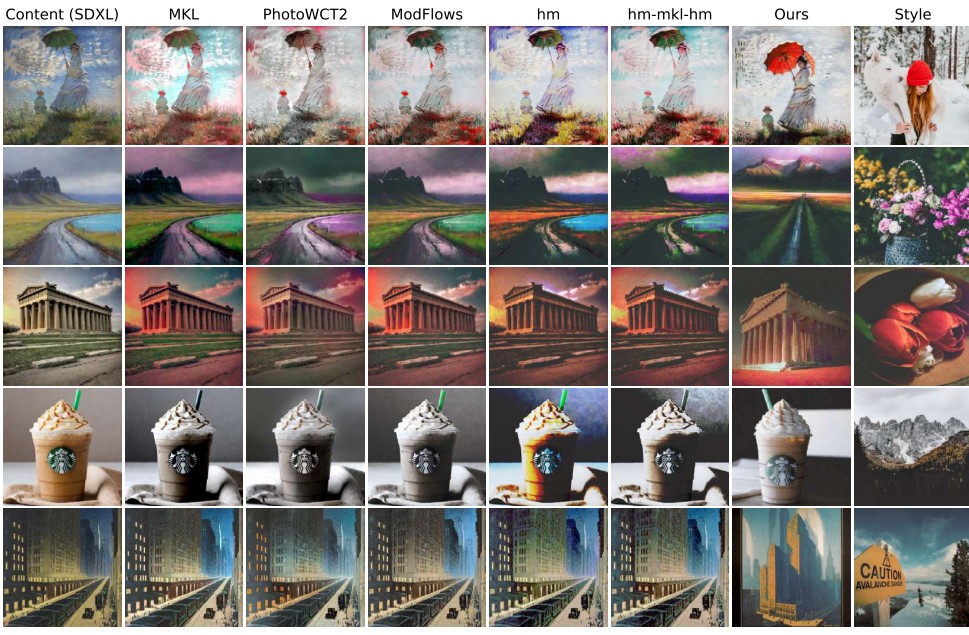

Figure 17: Text-to-image generation conditioned on a reference color distribution. Qualitative comparison with color transfer methods for SDXL. Please refer to the Table 1 in the main text for the quantitative comparison.

