# OpenReview forum: "Color Conditional Generation with Sliced Wasserstein Guidance"
_NeurIPS.cc/2025/Conference — NeurIPS 2025 spotlight_

### Official Review · Reviewer_rVkA · 2025-06-30

**Clarity:** 3
**Significance:** 4
**Originality:** 4
**Rating:** 5
**Confidence:** 4

**Summary:**

The paper proposes SW-Guidance, a training-free method for color-conditional image generation that modifies the sampling process of diffusion models using the differentiable Sliced Wasserstein distance to align the color distribution of generated images with that of a reference image. Unlike existing approaches that entangle style and color or rely on prompt engineering, SW-Guidance offers direct control over color without transferring unwanted textures or stylistic features. It achieves state-of-the-art performance in color similarity and semantic alignment with text prompts, as demonstrated through both quantitative metrics and qualitative results. The method is compatible with popular diffusion models like Stable Diffusion 1.5 and SDXL, and supports integration with structural controls like ControlNet.

**Questions:**

1. The method relies heavily on quantitative metrics such as 2-Wasserstein distance and CLIP-based scores. However, these may not fully align with human judgment, especially for color similarity. Could the authors include a small-scale human evaluation to validate whether their method's color outputs are perceptually preferred over baselines? Even a limited user study would provide valuable insight into real-world perceptual alignment.

2. The current experiments primarily focus on palettes with relatively uniform color characteristics. In practice, reference images often contain complex or mixed color themes. Can the authors provide either qualitative examples or additional results demonstrating SW-Guidance's robustness in cases where the reference image has multi-modal or ambiguous color distributions (e.g., natural scenes with mixed lighting or object diversity)?

**Ethical Concerns:**

["NO or VERY MINOR ethics concerns only"]

**Final Justification:**

The authors have adequately addressed all my concerns, including perceptual validation and inference analysis. The planned additions (e.g., qualitative results for complex palettes) further strengthen the submission. I maintain my initial score of Accept.

**Limitations:**

Yes

**Paper Formatting Concerns:**

I didn't find any major formatting issues.

**Quality:**

3

**Strengths And Weaknesses:**

Strengths

1. The paper presents a theoretically grounded, training-free method for integrating Sliced Wasserstein distance into the diffusion sampling process. This cleanly isolates color as a conditioning target without affecting texture or structure, representing a novel and modular extension to guidance methods.

2. Through comprehensive experiments on both SD1.5 and SDXL, the authors demonstrate superior performance in color fidelity, text alignment (CLIP-T), and visual quality. The inclusion of both stylization and color transfer baselines, along with ablation studies, supports the robustness of the approach.

3. The method is fully training-free, easy to plug into existing diffusion pipelines, and supported by open-source code. Its compatibility with structural guidance (e.g., ControlNet) enhances its applicability to creative and controllable image generation scenarios.

Weakness

1. While the paper reports strong quantitative results, it does not provide any human studies or perceptual assessments. This limits insight into real-world alignment with human visual preferences, especially for color.

2. The approach is evaluated on global color distributions, but does not explore real-world challenges such as multi-region palettes, spatial color layouts, or robustness to diverse lighting. This narrows its demonstrated applicability in practical design workflows.

3. Although the paper mentions inference time, it lacks detailed analysis of gradient computation overhead (e.g., memory, runtime scaling with image size or slice count), which could impact adoption in latency-sensitive or large-scale settings.

---

> ### Author Rebuttal · Authors · 2025-07-31
>
> We sincerely thank Reviewer rVkA for the positive feedback.
>
> **On Human Studies and Perceptual Assessment**  As we have noted in our responses to other reviewers, we believe a human perceptual study is a critical next step for validating our work. We conduct a human evaluation and will add it to the final version of the paper.  Although we had limited time and could not conduct a large-scale user study, we designed a small experiment in which each participant was asked to vote for the image that best matched a given color style. Each participant rated 10 image sets:
> | Rank | Method               | Votes) |
> |------|----------------------|---------------------|
> | 1    | SW-Guidance   | 60                |
> | 2    | InstantStyle  | 15              |
> | 3    |  IP-Adapter    | 8                 |
> | 4    |  RB Modulation | 5                 |
>
>
> **On Robustness to Complex Color Distributions** Please refer to Figure 4, that demonstrates applicability to the sharp palette. While a full quantitative analysis of these cases is extensive, we will add a new qualitative section to the appendix of our revised paper, showcasing SW-Guidance's performance multi-modal or ambiguous color distributions.
> We also agree that region-specific color control is an interesting and adjacent problem, as noted by the reviewer. One could apply SW-Guidance for a specific region of the generated image, the only adjustment which is needed is the computations of several SW distances for every region and reference color distribution.
>
>  **Inference Overhead** We thank the reviewer for raising these important points. We will expand our analysis to provide a more detailed breakdown of the computational overhead, as promised to other reviewers, in addition to ablation study in Fig 7. For a standard SDXL generation used a peak of **13301 MiB** of VRAM on our evaluation hardware, whereas generation with SW-Guidance used **15641 MiB**.
> Furthermore, we explored ways to mitigate overhead by avoiding backpropagation through the expensive U-Net. One potential direction we investigated involves leveraging the VAE decoder directly on the noised latent z_t at an intermediate timestep. While decoding a noised latent does not produce a semantically meaningful image, our findings suggest it still preserves some coarse color information to provide a useful guidance signal. However our preliminary study in this direction showed that in this case guidance effectively applied only on the later generation steps, which reduces color similarity and coherence of the generated image.

---

> > ### Comment · Reviewer_rVkA · 2025-08-05
> >
> > Thanks for the detailed response. The user study, memory breakdown, and ablation insights address my concerns well. I look forward to the added qualitative results on complex color distributions in the final version. No further questions from my side.

---

### Official Review · Reviewer_88qU · 2025-07-02

**Clarity:** 3
**Significance:** 3
**Originality:** 3
**Rating:** 4
**Confidence:** 3

**Summary:**

This paper proposes SW-Guidance, a training-free method for color-conditional image generation that modifies the sampling process of diffusion models using Sliced 1-Wasserstein distance to match reference color distributions. The approach demonstrates superior color matching while maintaining semantic coherence with text prompts compared to existing color transfer and stylized generation methods. The authors provide theoretical justification through Proposition 1 and Lemma 2, and extensive experiments validate their claims across SD-1.5 and SDXL architectures.

**Questions:**

1.While the paper provides extensive quantitative evaluation, the absence of human perceptual studies (e.g., preference tests or perceptual color matching assessments) leaves open questions about subjective quality and practical utility compared to existing methods.

2. The evaluation omits comparison with several recent state-of-the-art approaches in style transfer and prompt-aware stylization, making it difficult to fully assess the method's relative advantages.
[1]. SCSA: A Plug-and-Play Semantic Continuous-Sparse Attention for Arbitrary Semantic Style Transfer

3. While the paper evaluates traditional color transfer techniques (e.g., WCT2, PhotoWCT2) and diffusion-based stylization (e.g., IP-Adapter, InstantStyle), it omits critical discussion of ​​attention-based image editing methods​​ that achieve style alignment via feature adaptation or attention manipulation. A deeper comparative analysis would strengthen the paper by addressing:
[1] Unveil Inversion and Invariance in Flow Transformer for Versatile Image Editing.
[2] PS-Diffusion: Photorealistic Subject-Driven Image Editing with Disentangled Control and Attention.

4. The caption text in Figure 4 appears noticeably blurred, which compromises the figure's effectiveness.

5. typos issues in line 200.

**Ethical Concerns:**

["NO or VERY MINOR ethics concerns only"]

**Final Justification:**

The authors have addressed part of my concerns. I raise my initial score to borderline Accept.

**Limitations:**

yes

**Quality:**

3

**Strengths And Weaknesses:**

Strengths：
This work incorporates differentiable Sliced Wasserstein distance into diffusion model conditioning, providing a mathematically grounded solution to color-specific generation. The evaluation includes rigorous quantitative comparisons, showing statistically significant improvements. The method demonstrates compatibility with ControlNets and architectural agnosticism, working with both SD-1.5 and SDXL. The mathematical foundations are well-justified, with complete proofs provided in the Appendix.

Weaknesses:
1.While the paper provides extensive quantitative evaluation, the absence of human perceptual studies (e.g., preference tests or perceptual color matching assessments) leaves open questions about subjective quality and practical utility compared to existing methods.

2. The evaluation omits comparison with several recent state-of-the-art approaches in style transfer and prompt-aware stylization, making it difficult to fully assess the method's relative advantages.
[1]. SCSA: A Plug-and-Play Semantic Continuous-Sparse Attention for Arbitrary Semantic Style Transfer

3. While the paper evaluates traditional color transfer techniques (e.g., WCT2, PhotoWCT2) and diffusion-based stylization (e.g., IP-Adapter, InstantStyle), it omits critical discussion of ​​attention-based image editing methods​​ that achieve style alignment via feature adaptation or attention manipulation. A deeper comparative analysis would strengthen the paper by addressing:
[1] Unveil Inversion and Invariance in Flow Transformer for Versatile Image Editing.
[2] PS-Diffusion: Photorealistic Subject-Driven Image Editing with Disentangled Control and Attention.

---

> ### Author Rebuttal · Authors · 2025-07-31
>
> We thank Reviewer 88qU for the review. We understand the concerns raised, and we appreciate the opportunity to address them directly.
>
>  **On the Absence of Human Perceptual Studies**  To address this gap, we conduct a human evaluation study for the final version of the paper. Although we had limited time and could not conduct a large-scale user study, we designed a small experiment in which each participant was asked to vote for the image that best matched a given color style and textual description. Each participant rated 10 image sets :
> | Rank | Method               | Votes) |
> |------|----------------------|---------------------|
> | 1    | SW-Guidance   | 60                |
> | 2    | InstantStyle  | 15              |
> | 3    |  IP-Adapter    | 8                 |
> | 4    |  RB Modulation | 5                 |
>
>
>
> **On Missing Recent SOTA Comparisons** We will update our related work section to include a detailed discussion of SCSA, as well as the attention-based and flow-transformer editing methods suggested.
> We note that while these methods are powerful for general style transfer or subject-driven editing, they often operate by manipulating internal features and are not explicitly designed to match a specific global color distribution with the precision that SW-Guidance offers. Our approach is distinct in its objective function, which directly minimizes the distance between color distributions in the pixel space, offering a different and more direct form of control.
>
> **On Technical and Presentation Issues** We sincerely apologize for blurred images and other oversights. The blurred figure was an unfortunate artifact of compressing our figures to meet the submission file size restrictions.

---

### Official Review · Reviewer_sChs · 2025-07-02

**Clarity:** 4
**Significance:** 2
**Originality:** 2
**Rating:** 4
**Confidence:** 2

**Summary:**

The paper introduces SW-Guidance, a training-free control technique that steers a text-to-image diffusion sampler so that the generated image’s color distribution matches a given reference distribution provided as image. Building on Universal Diffusion Guidance, the key idea is to use at denoising steps a gradient signal from the Sliced 1-Wasserstein (SW) distance between the reference image and the current prediction.  Experiments are conducted on Stable Diffusion 1.5 and SD-XL and a subset of the ContraStyles dataset. The introduced method is compared to other histogram matching algorithms as well as established methods such as IP-Adapter and InstantStyle. Additionally, the authors showcase the compatibility of the introduced method in combination with ControlNet, introducing further (non-color) conditioning such as edge maps.

**Questions:**

- You note the need to differentiate through the UNet as a limitation. Have you experimented with rectified flow transformers such as Flux? Is Universal Guidance, and therefore your proposed approach, directly transferable to other architectures?

**Ethical Concerns:**

["NO or VERY MINOR ethics concerns only"]

**Final Justification:**

The clarification regarding architectural limitations, the discussion of applicability beyond color conditioning, and the promised additional analysis on inference overhead are appreciated and would strengthen the paper. However, as no concrete evidence or results have been provided for any of these points, I will maintain my score of 4 (borderline accept).

**Limitations:**

Yes, limitations are adequately addressed

**Quality:**

3

**Strengths And Weaknesses:**

**Strengths**

- While lightweight in the sense that no training or additional modules are required, the introduced method outperforms current approaches.
- The method is well introduced, and the paper is generally well written.
- Extensive experiments, including a wide range of baselines.
- Thorough ablations of hyperparameters and analysis of factors such as generation time.
- Limitations are well discussed.
- One of the limitations of the proposed approach is that the color steering alters the image content compared to unconditioned generation. However, the paper demonstrates that the method is compatible with ControlNets, which may mitigate this limitation.
- Publicly available code.

**Weaknesses**

- Compatibility with ControlNets is only demonstrated qualitatively.
- Inference time overhead is relatively high.
- The application appears to be quite niche, and the technical contribution is limited. Essentially, the proposed method combines Universal Guidance with the Sliced 1-Wasserstein distance.
 - The authors only consider UNet-based diffusion models and do not explore diffusion transformers or rectified flow transformers.

---

> ### Author Rebuttal · Authors · 2025-07-31
>
> We thank Reviewer sChs for the feedback.
>
>
>  **On Niche Application** Our method can be applied to other generation problems conditioned on a distribution. An example of such a task is text-to-speech generation conditioned on a person’s voice. Audio encoders WavLM [2] and FACodec [3] encode patches of 20 ms, producing a distribution of speaker embeddings from a long recording. This distribution captures richer characteristics of the speaker’s voice than a single embedding.
>
> While we build upon existing ideas, we propose using the differentiable Sliced Wasserstein distance for guidance in diffusion models. This is a non-trivial adaptation that requires careful theoretical and empirical validation. Our work provides a practical, training-free solution that effectively isolates color, a common and important creative need.
>
> [2] S. Chen et al., arXiv:2110.13900.
>
> [3] Z. Ju et al., arXiv:2403.03100.
>
>  **Inference Overhead** We thank the reviewer for raising these important points. We will expand our analysis to provide a more detailed breakdown of the computational overhead, as promised to other reviewers, in addition to ablation study in Fig 7. For a standard SDXL generation used a peak of **13301 MiB** of VRAM on our evaluation hardware, whereas generation with SW-Guidance used **15641 MiB**.
>
> Furthermore, we explored ways to mitigate overhead by avoiding backpropagation through the expensive U-Net. One potential direction we investigated involves leveraging the VAE decoder directly on the noised latent z_t at an intermediate timestep. While decoding a noised latent does not produce a semantically meaningful image, our findings suggest it still preserves some coarse color information to provide a useful guidance signal. However our preliminary study in this direction showed that in this case guidance effectively applied only on the later generation steps, which reduces color similarity and coherence of the generated image.
>
> **On Architecture Limitations** Regarding architectural generalizability, our method is not fundamentally limited to U-Nets. The core principle of our guidance is agnostic to the underlying network architecture. It requires only two conditions:
> 1.  A differentiable image prediction process.
> 2.  The ability to inject a guidance signal into the sampling loop.
>
> These conditions are met by most modern generative architectures, including Diffusion Transformers (DiT) and Rectified Flow models (e.g., Flux). In all these cases, our SW-Guidance loss would be applied to the model's prediction of the clean image (`x_0`), making the adaptation straightforward.

---

> > ### Comment · Reviewer_sChs · 2025-08-05
> >
> > Thank you for the response. The text-to-speech application appears reasonable. However, the paper would be significantly strengthened by empirical evidence demonstrating the generalization of the proposed approach beyond the color conditioning setting. I appreciate the details provided regarding memory consumption and look forward to the promised analysis of computational overhead. In addition to demonstrating empirical evidence on other architectures, it would be particularly insightful to examine how the computational overhead behaves in those contexts.

---

### Official Review · Reviewer_B7v7 · 2025-07-03

**Clarity:** 3
**Significance:** 3
**Originality:** 4
**Rating:** 5
**Confidence:** 4

**Summary:**

Color Conditional Generation with Sliced Wasserstein Guidance adds a training free technique to generative (text--to-image) diffusion models to control the color palette of the generation using a reference image. Using a differentiable version of the general form of the Sliced Wasserstein distance to condition the diffusion process style or content are successfully disentangled from color. Specifically, the Diffusion Posterior Sampling scheme as formulated in prior works Universal Diffusion Guidance and FreeDoM is used to incorporate the Sliced Wasserstein distance as a measure between two probability distributions into the conditioned diffusion process. The probability distributions correspond to the color distributions of generated image and color reference. While the regular prompt-based conditioning is still applied in latent space, the color guidance is added on the decoded image. The 1-Sliced Wasserstein distance is used as it is computationally lighter, implemented as a random 1D projection and L1 distance between the CDFs.
The method is evaluated on Stable Diffusion (SD) 1.5 and SDXL. with prompts from the ContraStyles dataset and references from Unsplash Lite. Qualitative results look promising with a superior disentanglement of color and style compared to other methods. Quantitative measurements look favorable for SD 1.5 and a little bit worse on SDXL with state-of-the-art performance in the distribution metric.

**Questions:**

- It seems like the method might be difficult to scale to newer diffusion architectures. Would it also work with e.g. a recitifed flow model? What are the adjustments necessary?
- How do papers like [1, 2]  relate to the claim that textual input is impractical for color control? It seems like adding RGB tuples to text prompts works for e.g. StableDiffusion 1.4. Adding [2] to the related works section would make sense, I think.
- What do the results of the CLIP-T in Table 1 really mean? The text mentions the highest score for the SW Guidance which doesn't seem to be the case in the table here. A bit more discussion of the CLIP results could be interesting here.
- What is happening in l.115? The sentence seems broken.
- l. 185, 186: Isn't a good performance on the 2-Wasserstein distance expected after optimization for the 1-Wasserstein distance?

1: Bordin et al. Fine color guidance in diffusion models and its application to image compression at extremely low bitrates. ArXiv 2024.
2: Butt et al. ColorPeel: Color Prompt Learning with Diffusion Models via Color and Shape Disentanglement. ECCV 2024

**Ethical Concerns:**

["NO or VERY MINOR ethics concerns only"]

**Final Justification:**

With the corrections and improvements made that the authors announced in the rebuttal, the paper would be a relevant addition to the conference program. The proposed perceptual study certainly is a good addition.
Still the scope of the work remains limited due to the paper mostly concentrating on Unet-based diffusion models and the need to backpropagate through the network. Therefore, my recommendation to accept the paper still holds but is not increased to a strong accept.

**Limitations:**

Yes.

**Paper Formatting Concerns:**

No major concerns. The footnote in l. 53 should be added after the punctuation maybe.

**Quality:**

3

**Strengths And Weaknesses:**

Strengths:
- The combination of the Sliced Wasserstein Distance, mostly known from traditional statistical image optimization tasks with the generative image diffusion paradigm is an original and novel idea.
- Nice documentation of the method's limitations.
- The paper is overall nicely written. Interesting choice to incorporate the related works fully into the introduction (which is somewhat uncommon for computer science papers these days) but it works well.
- The method seems flexible and can, for example, be combined with the ControlNet architecture, as shown.

Weaknesses:
- The method needs to backpropagate through the entire U-Net of the diffusion model. Especially for larger (more recent) diffusion networks, this circumstance will make the approach infeasible. Showing a way to overcome this limitation would give this paper a significantly higher impact, I believe.

Overall, the paper presents an original idea paired with a useful application. There are some minor questions left (see questions), but the method could be a good fit for the NeurIPS conference.

---

> ### Author Rebuttal · Authors · 2025-07-31
>
> We sincerely thank Reviewer B7v7 for the positive evaluation.
>
> **Generality** We note that our guidance mechanism is architecturally agnostic and works with any generative model that is differentiable. For rectified flow models, the adaptation would be straightforward: our guidance term would be applied to the predicted clean data, `x_0`, in the same manner as it is in our current DDIM-based implementation.
>
> **How do papers like Bordin et al. and Butt et al. relate to the claim that text is impractical for color control?**
> Thank you for pointing out these relevant works; we will add them to our related work section. We agree that text-based color control has its uses, but our claim is that it lacks the precision and fidelity required for matching complex color palettes. Its primary limitations, which our method overcomes, are:
> -   **Limited Precision:** Text is often insufficient for describing a full, nuanced color distribution.
> -   **Difficulty Specifying Distributions:** Text struggles to specify the relative amounts and interplay of multiple colors.
>
> While the approach by Bordin et al. is interesting, it focuses on spatially-aware color control (e.g., placing specific colors in specific regions), which is a different goal from our objective of matching a *global* color distribution and limits diversity of generation. In the paper we also discuss and compare to the similar approach of guiding with spatial color information [1], please refer to the Table 4 in Supplementary.
>
> [1] ghoskno. Color-canny controlnet. https://huggingface.co/datasets/ghoskno/ laion-art-en-colorcanny, 2023.
>
> **Results of the CLIP-T** Thank you for highlighting the lack of clarity; we will revise the text to be more precise. Our claim is that SW-Guidance achieves the highest CLIP-T score among methods that perform color/style conditioning, such as IP-Adapter, RB-Modulation and InstantStyle.
> | Method                         | CLIP-T Score ↑       |
> |--------------------------------|-----------------------|
> | SW-Guidance SDXL (Ours)        | 0.270 ± 0.002         |
> | RB-Modulation StableCascade    | 0.266 ± 0.003         |
> | InstantStyle SDXL              | 0.238 ± 0.002         |
> | IP-Adapter SDXL                | 0.214 ± 0.002         |
>
> **l.115** Thank you for catching this. We will correct this typo in the revised manuscript, which is supposed to be a discussion of the algorithm pictured in Fig2.
>
> **Isn't a good performance on the 2-Wasserstein distance expected after optimization for the 1-Wasserstein distance?** We use 2-Wasserstein as a standard evaluation metric in the field.  We will clarify this relationship in the paper.

---

> > ### Comment · Reviewer_B7v7 · 2025-08-05
> >
> > Given the proposed revisions and clarifications from the rebuttal I don't have further questions and already updated my final justification accordingly. Overall, I see the paper as a suitable addition to the conference but some limitations in scope remain that prevent me from awarding the highest score. Let me know if there are any further questions.

---

### Official Review · Reviewer_SYqP · 2025-07-03

**Clarity:** 3
**Significance:** 2
**Originality:** 3
**Rating:** 5
**Confidence:** 3

**Summary:**

The authors propose SW-Guidance, a training-free and optimization/guidance-based method for color distribution conditioned image generation with T2I diffusion models. Particularly, since the color distribution of an image is defined by a CDF, the authors use the sliced Wasserstein distance to compare the color CDFs of the generated and reference images, defining a loss which is optimized w.r.t. the latent for every denoising step similar to Universal Guidance.

**Questions:**

1. Why was the 2-Wasserstein distance between CDFs chosen as the quantitative evaluation metric? Since the method directly optimizes 1-Wasserstein distance, it can seem like the method is directly optimizing for the quantitative evaluation metric, which makes it less good of a metric.
2. From the examples shown in Figure 13 and 17, it seems like SW-Guidance slightly changes the content and structure of the generated image compared to vanilla T2I generation from the same seed, which makes sense for guidance. However, does SW-Guidance limit the diversity of T2I generation, where the color palette may encourage the output image to have a certain style, structure, or texture? I would love to see the authors investigate this possibly with FID (lower means more diverse generation) as this seems to echo the quality/category adherence and diversity tradeoff of classifier-free guidance. To possibly reduce effects of matching color distributions on FID, the authors can evaluate images after they have been converted to grayscale and/or passed through histogram normalization.
3. These two questions are purely for discussion, but how does having to differentiate through the SDXL VAE affect the method’s inference time and VRAM usage, and would it be possible to find some way to compute color CDFs in the VAE latent space? Moreover, what are some other conditional generation applications/objectives the sliced Wasserstein guidance approach can be applied to? This can help illuminate the significance of the work.

Addressing weaknesses 1 and 2, clarifying question 1, and providing some experiments and discussions on question 2 can push my score from a 4 to a 5. Discussions on question 3 would be well-appreciated too!

**Ethical Concerns:**

["NO or VERY MINOR ethics concerns only"]

**Final Justification:**

The authors have addressed most of my questions and concerns, and I especially appreciate the added experiment to show how SW-Guidance is able to preserve output diversity compared to existing baselines. I urge the authors to incorporate the rebuttal into the final version of the paper. After reading other reviewers’ comments and the authors’ responses, I raise my final rating to Accept.

**Limitations:**

Yes.

**Quality:**

3

**Strengths And Weaknesses:**

1. It would be great if the authors acknowledged existing optimization/guidance-based methods for T2I diffusion controllable generation, such as Diffusion Self-Guidance [1] and FreeControl [2], especially since both works enable scene appearance control which is adjacent to color control.
2. It would also be great to see SW-Guidance compared to naive approaches to color transfer/stylization with a baseline of a simple loss function such as the following, which can more visually show the advantages of SW-Guidance’s loss function design: 1) L2 loss between the mean and standard deviation of intermediate diffusion features (which these two aforementioned methods use), or 2) L1/L2 loss between the CDFs of the generated and reference images without random rotations which come from the sliced Wasserstein distance.
3. This is more of a suggestion: Algorithm 2 is essentially a much more detailed version of Algorithm 1. It would be nice to acknowledge Algorithm 2 when introducing Algorithm 1 as otherwise one has to dig into the appendix to realize that there is a more detailed version of the proposed method.

[1] Dave Epstein, Allan Jabri, Ben Poole, Alexei A. Efros, Aleksander Holynski. “Diffusion Self-Guidance for Controllable Image Generation.” NeurIPS 2023, https://arxiv.org/abs/2306.00986.

[2] Sicheng Mo, Fangzhou Mu, Kuan Heng Lin, Yanli Liu, Bochen Guan, Yin Li, Bolei Zhou. “FreeControl: Training-Free Spatial Control of Any Text-to-Image Diffusion Model with Any Condition.” CVPR 2024, https://arxiv.org/abs/2312.07536.

---

> ### Author Rebuttal · Authors · 2025-07-31
>
> We thank Reviewer SYqP for the thoughtful feedback and constructive suggestions.
>
>
> **Acknowledgment of existing optimization/guidance-based methods like Diffusion Self-Guidance and FreeControl.** We appreciate this suggestion and agree that discussing these relevant works will better contextualize our paper. We note that while these methods excel at providing spatial and structural control, our work specifically targets the matching of an image's global color distribution, which is a distinct goal within the broader field of controllable generation. Also it is worth noting that color information is entangled with spatial features within attention maps, which makes it hard to control only color of the generated image without affecting other features.
>
> **On comparisons with other loss functions.** This is an excellent suggestion that will strengthen our ablation study. The SW distance and its variations guarantee weak convergence of the generated color distribution to the reference one. There is no such property for L2 loss on mean/std and L1/L2 loss on CDFs without random rotations.
> We acknowledge that our paper did not consider attention-based or feature-map-based methods and thus we defer this investigation of applying loss on features for later investigation. Regarding showing that sliced wasserstein performs better we refer to ablation study, we compare our method with matching first moments of distributions.
> **Algorithm Presentation** We will add a reference to the more detailed Algorithm 2 from the appendix when we first introduce the main algorithm in the body of the paper.
>
> **Why was the 2-Wasserstein distance chosen for evaluation when the method optimizes for the 1-Wasserstein distance?**. We chose the 2-Wasserstein distance as it is a standard and widely used metric in the color transfer literature and in the broader field of density matching. However, to provide a more comprehensive and unbiased evaluation, we will expand our analysis to include additional metrics. As suggested by the reviewer in a later question, we will add FID scores.
> **Does SW-Guidance limit the diversity of T2I generation?** To provide a quantitative answer, we have followed the reviewer's suggestion and computed the FID scores between unconditional SDXL generations and those from various style guidance methods. The results, calculated on our generated dataset, are as follows:
>
> | Method Used with SDXL | FID Score (vs. Unconditional) |
> | :--- | :--- |
> | Mean/Covariance Matching Only | 53.16 |
> | **SW-Guidance (Ours)** | **58.40** |
> | InstantStyle | 58.95 |
> | IP-Adapter | 71.06 |
> | RB Modulation | 72.75 |
>
> They show that SW-Guidance maintains content diversity on par with other state-of-the-art stylization methods. While there is a slight increase in FID compared to simple moment matching (which provides weaker color control), our method is significantly better at preserving diversity than stronger stylization methods like IP-Adapter and RB Modulation.
>
> **What is the computational overhead, and could CDFs be computed in latent space?**
> This is an important practical consideration. We will add a more detailed analysis of the computational overhead in addition to ablation in Fig 7. For a standard SDXL generation used a peak of **13301 MiB** of VRAM on our evaluation hardware, whereas generation with SW-Guidance used **15641 MiB**.
>
> Regarding the use of latent-space CDFs, we agree this is an interesting direction. As color is an explicit property of the pixel space, computing exact CDFs requires decoding the image. However, one could explore lightweight, learnable models that approximate color statistics from latent features, which could improve efficiency in future work.
> Furthermore, we explored ways to mitigate overhead by avoiding backpropagation through the expensive U-Net. One potential direction we investigated involves leveraging the VAE decoder directly on the noised latent z_t at an intermediate timestep. While decoding a noised latent does not produce a semantically meaningful image, our findings suggest it still preserves some coarse color information to provide a useful guidance signal. However our preliminary study in this direction showed that in this case guidance effectively applied only on the later generation steps, which reduces color similarity and coherence of the generated image.

---

### Decision · Program_Chairs · 2025-09-17

**Decision:**

Accept (spotlight)

**Comment:**

This paper introduces SW-Guidance, a training-free method for controlling the color palette of images generated by diffusion models. The main idea is to guide the sampling process with a loss based on the Sliced 1-Wasserstein distance, which directly compares the color distribution of the generated image to that of a reference palette. The authors claim this technique effectively disentangles color from content, producing images that match the reference colors while staying true to the text prompt.

The paper's primary strengths are its novelty, strong performance, and thorough evaluation. The main idea of using Sliced Wasserstein distance for guidance was noted as original and theoretically sound (B7v7, rVkA). It is a lightweight, training free method, and outperforms more complex baselines in color matching while maintaining text coherence (sChs, rVkA). This was supported by extensive experiments and clear ablations (sChs, SYqP, rVkA), and the paper was consistently described as well-written and easy to follow (B7v7, sChs).

Initially, reviewers' concerns were less about the method's core function and more about its context and practical limits. Reviewers noted gaps in the evaluation, e.g., human perceptual studies (88qU, rVkA) and comparison against a broader set of controllable generation methods (SYqP, 88qU). The most significant technical weakness raised was scalability and overhead, as the required backpropagation through the entire U-Net is costly (B7v7, sChs).

The rebuttal was strong and resolved many issues. Still, the method's scalability and computational overhead remains the main unresolved weakness. The need to backpropagate through the entire U-Net is a limitation for future, larger architectures (B7v7, sChs).

Overall, the AC finds that this paper to introduce a novel, elegant, and effective training free method for color control that earned a strong consensus for acceptance from all five reviewers. While the computational overhead is a fair limitation on future scalability, it does not undermine the significance of the present work. A spotlight is recommended as the core idea of using Sliced Wasserstein guidance is a clever and principled technique with the potential to inspire similar distributional control methods in other domains, making it a work of broader interest.